# Visualizing stepwise evolution of carbon hybridization from *sp³* to *sp²* and to *sp*

Wei Xiong[1,8], Guang Zhang[2,8], De-Liang Bao [ORCID][3,8], Jianchen Lu [ORCID][1,8] ✉, Lei Gao [ORCID][4], Yusen Li[2], Hui Zhang[1], Zilin Ruan [ORCID][1], Zhenliang Hao[1], Hong-Jun Gao [ORCID][5], Long Chen [ORCID][2,6] ✉ & Jinming Cai [ORCID][1,7] ✉

Regulating carbon hybridization states lies at the heart of engineering carbon materials with tailored properties but orchestrating the sequential transition across three states has remained elusive. Here, we visiualize stepwise evolution in carbon hybridizations from *sp³* to *sp²* and to *sp* states via dehydrogenation and elimination reactions of methylcyano-functionalized molecules on surfaces. Utilizing scanning probing microscopy, we distinguish three distinct carbon-carbon bond types within polymers induced by annealing at elevated temperatures. Density-functional-theory calculations unveil the pivotal role of the electron-withdrawing cyano group in activating neighboring methylene to form C(*sp³*)–C(*sp³*) bonds, and in facilitating subsequent stepwise HCN eliminations to realize the transformation across three carbon-carbon bond types. We also demonstrate the applicability of this strategy on one-dimensional molecular wires and two-dimensional covalent organic framework on different substrates. Our work expands the scope of carbon hybridization evolution and serves as an advance in flexibly engineering carbon-material by employing cyanomethyl-substituted molecules.

The topographies and properties of carbon-based nanomaterials markedly depend on the hybridization of the bonding carbon atoms ($sp^n$, $n = 1$, 2, or 3)[1]. When $n$ changes from 1, 2, to 3, the geometries of carbon-carbon bonds transform successively from three dimension (e.g., diamond) to two dimension (e.g., graphene) and finally to one dimension (e.g., polyacetylene), which brings developments in various fields of nanoelectronics, catalysis and photovoltaic device, *etc*[1-6]. Therefore, the regulation of the specific carbon hybridization is important for controlling the topology, dimensions, and morphology of targeted carbon materials, as well as imparting tailored physico-chemical properties. Recent efforts have enabled the creation and functionalization of carbon materials by altering carbon hybridization from one state to another[7-14]. Modulating the sequential transition

across all three states has remained an elusive yet highly desirable capability, promising higher flexibility in sculpting carbon materials with finely tuned properties[1,15].

The evolution of carbon hybridization is prevalent during the preparation and modification of multifunctional carbon-based materials via traditional wet chemistry[16-19]. However, challenging factors of capturing the intermediates in the reaction and tracing the evolving process of hybridization are insurmountable. Additionally, the catalysts and solvents employed in synthetic process may become potential interferences for the precise characterization of carbon hybridization states. By leveraging the capabilities of scanning tunneling microscopy (STM), bond-resolved scanning tunneling microscopy (BR-STM) and non-contact atomic force microscopy (nc-AFM),

[1]Faculty of Materials Science and Engineering, Kunming University of Science and Technology, 650093 Kunming, PR China. [2]Department of Chemistry and Tianjin Key Laboratory of Molecular Optoelectronic Science, Tianjin University, 300072 Tianjin, PR China. [3]Department of Physics and Astronomy, Vanderbilt University, Nashville, TN 37235, USA. [4]Faculty of Science, Kunming University of Science and Technology, 650093 Kunming, PR China. [5]Institute of Physics & University of Chinese Academy of Sciences, 100190 Beijing, PR China. [6]State Key Laboratory of Supramolecular Structure and Materials, College of Chemistry, Jilin University, 130012 Changchun, PR China. [7]Southwest United Graduate School, 650093 Kunming, PR China. [8]These authors contributed equally: Wei Xiong, Guang Zhang, De-Liang Bao, Jianchen Lu. ✉e-mail: jclu@kust.edu.cn; longchen@jlu.edu.cn; j.cai@kust.edu.cn

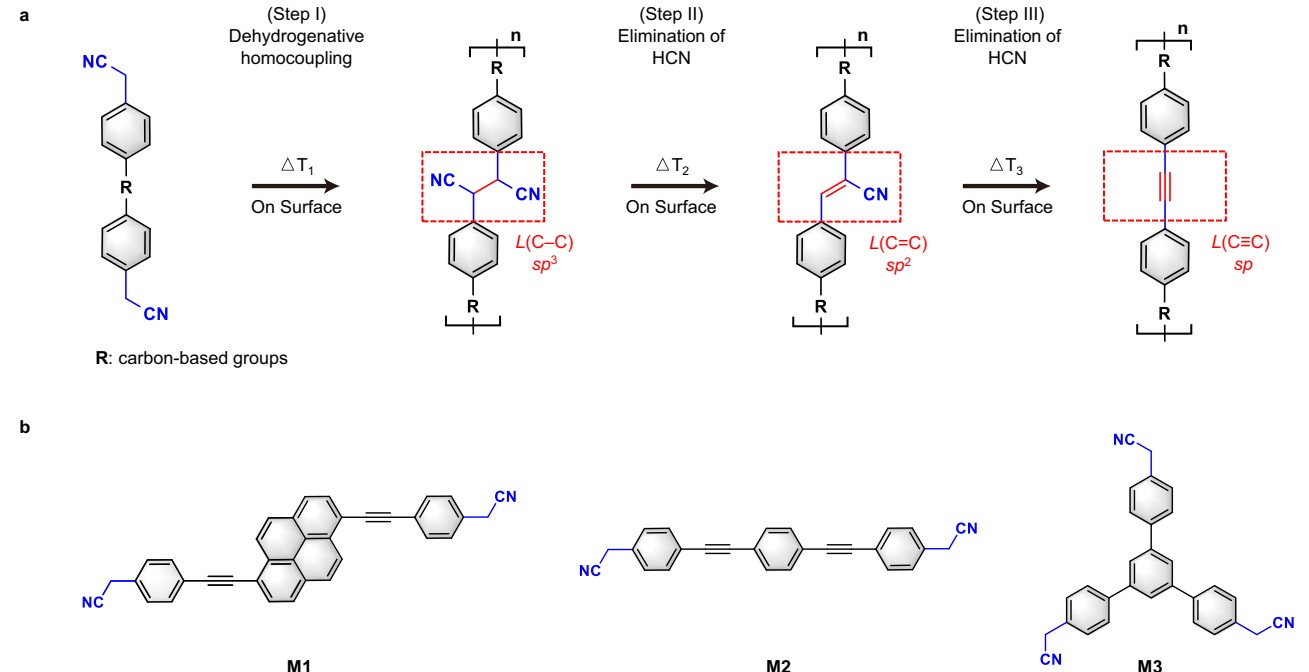

**Fig. 1 | Illustration of sequentially regulating carbon hybridization states by employing the (4-methylcyano)phenyl-substituted compounds on surface.**
**a** The dehydrogenative homocoupling (step I) is initiated at temperature $\Delta T_1$, forming the 1,2-dicyanoethyl-linked polymer with $sp^3$-hybridized linkages $L$(C–C). Thermal annealing to $\Delta T_2$ induces the elimination of HCN (step II), leading to the formation of cyanoethylene-linked polymer with $sp^2$-hybridized linkages $L$(C=C). Further annealing to elevated temperature $\Delta T_3$ triggers further elimination of HCN (step III), resulting in the ethynylene-linked polymer with $sp$-hybridized linkages $L$(C≡C). Red dashed boxes highlight the defined linkages. $\Delta T_1 < \Delta T_2 < \Delta T_3$. **b** Three prototype precursors employed in this work are labeled as M1, M2, and M3.

surface science has offered a versatile and convenient approach to regulating and characterizing distinct carbon hybridization states at atomic level via diverse strategies, such as on-surface chemical reactions, tip manipulations and photocycloaddition[7,8,14,20,21].

The first and pivotal step towards realization of carbon hybridization across three types involves the formation of C($sp^3$)–C($sp^3$) bonds. The on-surface dehalogenation reactions have garnered significant attention for its ability to form intermolecular carbon-carbon bonds in polymers and carbon nanostructures[10,13,22–25], while falling short in achieving the transition of the as-formed carbon-carbon bonds. An alternative way is through C–H activation[12,26]. In wet chemistry, it has been studied that methylene can be activated by adjacent electron-withdraw groups (EWGs), such as cyano (–CN). On the other hand, the –CN group is reported to engage in elimination reactions on surfaces by forming HCN[27]. In this work, we particularly design and fabricate precursors functionalized with –CH$_2$–CN groups and propose the reaction scheme as outlined with three steps in Fig. 1a. Step (I): The –CN groups facilitate dehydrogenative homocoupling between monomers, forming linkages of –CH(CN)–CH(CN)–, named as $L$(C–C), including C($sp^3$)–C($sp^3$) bonds (left red dashed box in Fig. 1a). Step (II): One –H and one –CN groups are eliminated from the linkage after thermal annealing, transforming the linkages to be –CH=C(CN)–, named as $L$(C=C), including C($sp^2$)=C($sp^2$) bonds (middle red dashed box in Fig. 1a). Step (III): The remaining –H and –CN groups at the linkage further eliminate, transforming the linkages to be –C≡C–, named as $L$(C≡C), including C($sp$)≡C($sp$) bonds (right red dashed box in Fig. 1a). Chemical structures of targeted species on metal surfaces are elucidated by STM, BR-STM and nc-AFM. The intrinsic mechanism is unveiled by means of density-functional-theory (DFT) calculations, which presents an explicit outline for the entire reactions.

We rationally designed and synthesized three prototype precursors, 1,6-di[2-(4-cyanomethylphenyl)ethynyl]pyrene (**M1**), 1,4-di[2-(4-cyanomethylphenyl)ethynyl]benzene (**M2**), and 2,2'-(5'-(4-(cyanomethyl)phenyl)-[1,1':3',1''-terphenyl]-4,4''-diyl)diacetonitrile (**M3**), as

seen in Fig. 1b (detailed synthetic routes are exhibited in supplementary information). The design motivation for three precursors stems from the following perspectives: (i) The –CH$_2$–CN groups within **M1, M2** and **M3** allow the formation and regulation of carbon-carbon bonds in 1D nonlinear polymer, 1D linear polymer and 2D COF, respectively. (ii) The intramolecular C($sp$)≡C($sp$) bonds of **M1** and **M2** enhance the planarity of the molecular skeleton, allowing for better surface characterization. (iii) The newly-formed C($sp$)≡C($sp$) bonds can be identified by comparison with the existing ones within the precursors **M1** and **M2** as the references.

## Results

### Dehydrogenative homocoupling of methylene activated by –CN groups

We firstly target the **M1**-based reaction on Au(111). Figure 2a shows the schematic reaction of dehydrogenative homocoupling of **M1**, illustrated as step (I) in Fig. 1a, leading to the formation of the linkage $L$(C–C) that connects two –CH$_2$– groups from two reacting **M1** molecules. The large-scale STM image (Fig. 2b) presents the self-assembled monomers with parallel arrangement before the polymerization. Zoom-in BR-STM (Fig. 2c) and nc-AFM (Fig. 2d) show chemical-bond resolved images, demonstrating excellent alignment with the superimposed chemical structures. The inter- and intra-molecular bonds are distinguished clearly (Fig. 2c). Due to the $sp^3$-hybridization of the carbon atom in the –CH$_2$– group, two H atoms connected to C($sp^3$) are oriented towards the substrate and the vacuum respectively, exhibiting a non-planar bonding geometry. This results in a reduced distance between the tip and the –CH$_2$–group, specifically manifesting as bright protrusions in nc-AFM images (white dotted circle Fig. 2d)[28]. The reaction is triggered upon annealing the self-assembled structure to 520 K for 20 min. Figure 2e shows a large-scale STM image of the as-formed polymer **1A**. The depressions that are regularly located at both sites of linkage within polymer **1A** (indicated by white arrows) are ascribed to local changes of the tunneling barrier via surface dipoles

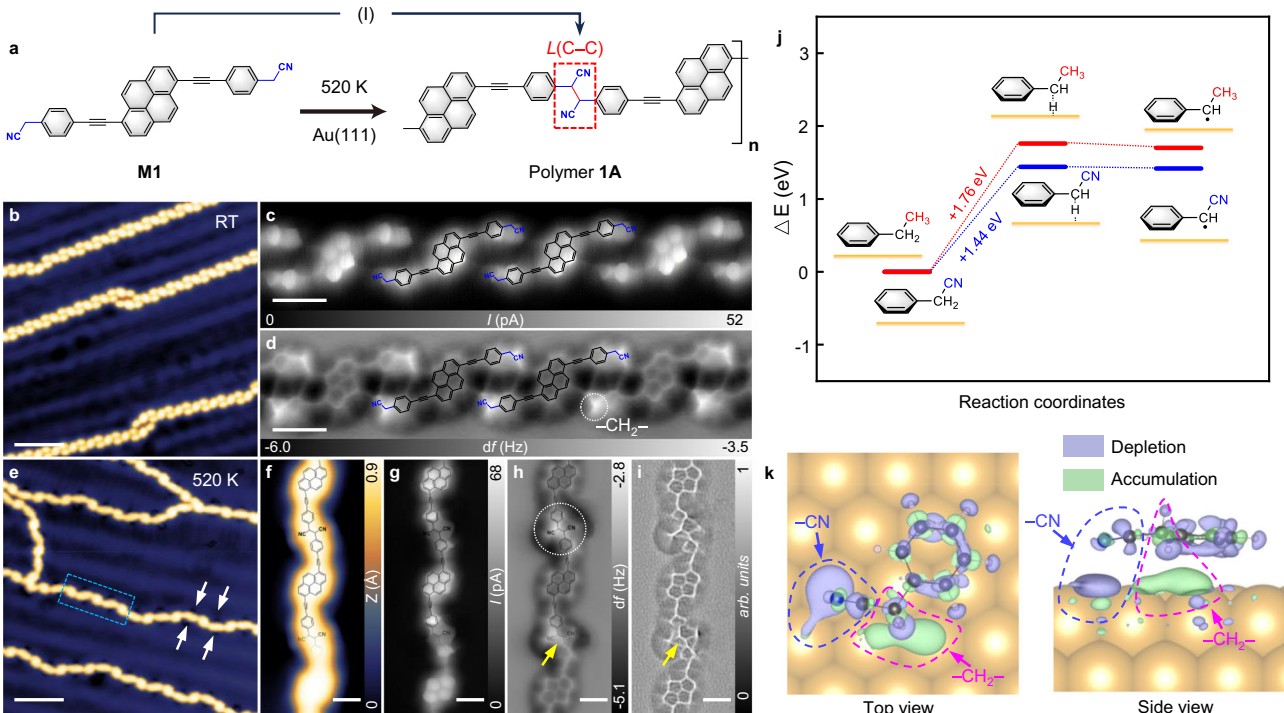

**Fig. 2 | Dehydrogenative polymerization of M1 on Au(111) surface. a** Schematic representation of step (I) of **M1**, dehydrogenative homocoupling to form polymer **1A** on Au(111). **b** Large-scale STM image (200 mV, 200 pA) showing self-assembled precursors **M1** after room temperature (RT) deposition onto Au(111). **c** Zoom-in BR-STM image (2 mV, tip-sample distance offset Δz = 180 pm) of self-assembled precursors **M1** on Au(111). **d** nc-AFM image (2 mV, Δz = 220 pm) corresponding to (**c**). The images of (**c**, **d**) are overlaid with the chemical structures of **M1**. **e** Large-scale STM image (200 mV, 100 pA) showing polymers **1A** after annealing to 520 K for 20 min. The depressions on both sides of linkages are marked by white solid arrows. **f**–**i** Zoom-in STM image (200 mV, 100 pA) (**f**), BR-STM image (2 mV, Δz = 170 pm) (**g**), nc-AFM image (2 mV, Δz = 200 pm) (**h**) and Laplace-filtered image (**i**) of polymer **1A** derived from the blue dashed box in (**e**). The images of (**f-h**) are overlaid with the proposed chemical structures of polymer **1A**. Yellow arrows in (**h**, **i**) highlight as-formed L(C–C). White dotted circle indicates the brighter contrast of L(C–C) characterized by nc-AFM imaging. **j** DFT-calculated energy barriers for methylene dehydrogenation in prototype molecules of ethylbenzene (higher) and phenylacetonitrile (lower). IS: initial state. TS: transition state. FS: final state. **k** Top and side views (left and right panels, respectively) of charge-density difference between the adsorbed phenylacetonitrile molecule and Au(111) substrate. Purple: electron depletion; Green: electron accumulation. Isosurface: $4.6 \times 10^{-3}$ e/Å³. DFT simulation functional: PBE + D3. Scale bars, 5 nm (**b**, **e**), 1 nm (**c**, **d**), 0.6 nm (**f–i**).

generated by the lone pair of –CN groups[29,30]. Zoom-in images in Fig. 2f-i unveil the structural details of polymer **1A**. The newly formed L(C–C) between molecules are clearly imaged (yellow arrows in Figs. 2h and 2i), contrasting to the non-intermolecular-bonding features in Figs. 2c and 2d. The $sp^3$-hybridized L(C–C) is evidenced by the brighter contrast in the nc-AFM image (white dotted circle in Fig. 2h), which is attributed to the non-planar tetrahedral configurations. The –CN groups exhibit a characteristic of terminal bifurcation in nc-AFM imaging, consistent with the results reported in previous work[31]. The covalent nature of polymer **1A** is confirmed by lateral tip manipulation experiments (Supplementary Fig. 1)[32]. Further zoom-in BR-STM and nc-AFM images of polymer **1A** in Supplementary Fig. 2 show more detailed imaging. The d$I$/d$V$ spectra and d$I$/d$V$ mappings acquired on the polymer **1A** present analogous electronic states of two neighboring linkages L(C–C), which attests to the homogeneity of the dehydrogenative polymerization (Supplementary Fig. 3). More samples obtained via stepwise annealing are displayed in Supplementary Figs. 4 and 5. Notably, the homocoupling of cyano-motivated –CH₂– groups at step (I) represents a significant breakthrough in ultra-highly selective dehydrogenation reaction of saturated C($sp^3$)–H bond (Supplementary Fig. 6).

We performed DFT calculations to demonstrate the dominant role of EWG –CN, in activating its neighboring methylene during step (I). We calculated the energy barriers of on-surface –CH₂– dehydrogenation with two prototype molecules, ethylbenzene and phenylacetonitrile, where the reacting –CH₂– groups connect with an electron-donating group (–CH₃) and an EWG (–CN), respectively (Fig. 2j). The simplified model of phenylacetonitrile is believed to reasonably describe the energy barriers (Supplementary Fig. 7). The calculated barrier for phenylacetonitrile (1.44 eV) is lower than that for ethylbenzene (1.76 eV), demonstrating that the electron-withdrawing effect of –CN activates C–H bonds in methylene groups. Calculated atomic models are exhibited in Supplementary Fig. 8. Additionally, we specifically designed the molecule 4,4'-diethylbiphenyl (**DBP**) to conduct a comparative analysis of the selectivity displayed in the dehydrogenation reactions of –CH₂– groups on Au(111) surface, as opposed to methylcyano-functionalized molecules. Our experimental results indicate that, unlike –CN groups, the terminal –CH₃ groups are unable to selectively activate adjacent –CH₂– groups (see Supplementary Fig. 9 for further details).

The substrate plays a role in selectivity during the C–H activation. Figure 2k shows the charge density difference between the phenylacetonitrile molecule and the substrate upon adsorption. The depletion of π electrons of the benzene ring indicates obvious charge redistribution between the molecule and the substrate. Furthermore, the charge redistributes non-uniformly at the interface. Near the –CN group (blue dashed curve in Fig. 2k), the electrons are depleted because of the electron-withdrawing effect of the –CN group. Instead, the electrons are accumulated beneath –CH₂– (pink dashed curve in Fig. 2k), forming a negatively charged region that selectively activates the near-substrate C–H bond for dehydrogenation. It results that the electron localization within the near-substrate C–H bond is weakened by the substrate, compared to that within the off-substrate C–H bond, as shown in Supplementary Fig. 10. Conversely, in the case of

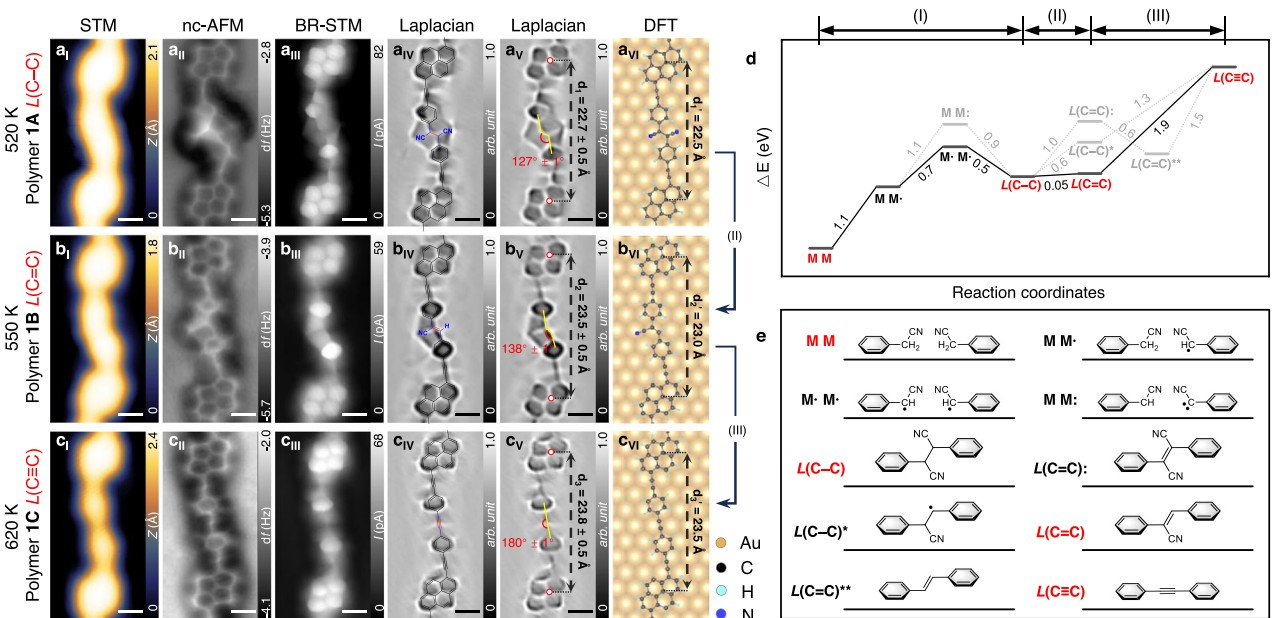

**Fig. 3 | Stepwise transition of carbon hybridization across all three states at the linkage via annealing to elevated temperatures.** $a_I$–$a_{VI}$ Polymer 1A with the linkage $L$(C−C). $b_I$–$b_{VI}$, Polymer 1B with the linkage $L$(C=C). $c_I$–$c_{VI}$ Polymer 1C with the linkage $L$(C≡C). STM images ($a_I$–$c_I$), nc-AFM images ($a_{II}$–$c_{II}$), BR-STM images ($a_{III}$–$c_{III}$), Laplace-filtered BR-STM images overlaid with chemical structures ($a_{IV}$–$c_{IV}$), Laplace-filtered BR-STM images ($a_V$–$c_V$) showing distinct bond angles and relative distances in three polymers, and DFT-optimized structures ($a_{VI}$–$c_{VI}$).

Scanning parameters: ($a_I$–$c_I$) 200 mV, 200 pA. ($a_{II}$–$c_{II}$) 2 mV, $\Delta z$ = 210 pm. ($a_{III}$–$c_{III}$) 2 mV, $\Delta z$ = 160 pm. **d** DFT-calculated relative energies for corresponding configurations. The solid line presents the most possible reacting pathway, in which the species highlighted in red are observed in experiments. While dashed lines present other considered paths. **e**, Illustrations of the configurations in (**d**). DFT simulation functional: PBE + D3. Scale bar, 0.4 nm ($a_I$–$c_V$).

ethylbenzene on Au(111), there is an insignificant charge redistribution observed between the −CH₂−CN groups and the substrate, suggesting no enhancement in the dehydrogenation process (Supplementary Fig. 11).

## The evolution of carbon hybridization across three states on gold surface

Further step-by-step annealing leads to stepwise eliminations of HCN species and thus transition of carbon hybridization at the linkages. STM (Fig. 3$a_I$-$c_I$), nc-AFM (Fig. 3$a_{II}$-$c_{II}$) and BR-STM (Fig. 3$a_{III}$-$c_{III}$) characterizations are employed to characterize the corresponding products. The polymer **1A** (Fig. 3$a_I$-$a_{VI}$) with linkage $L$(C−C) is regarded as a referenced species in the following annealing process. The length between two adjacent pyrene skeletons is measured to be $d_1 = 22.7 \pm 0.5$ Å (Fig. 3$a_V$), showing good agreement with the DFT-optimized value (22.5 Å) (Fig. 3$a_{VI}$).

Upon annealing to 550 K for 20 min, it is evident that the STM image (Fig. 3$b_I$) shows an asymmetric feature at the linkage within polymer **1B**, in contrast to the symmetric feature observed in polymer **1A** (Fig. 3$a_I$). Polymer **1B** notates the product of step (II) of **M1**, where one −H and one −CN groups are eliminated from the two C atoms at the linkage, resulting to $L$(C=C), namely, −CH=C(CN)− group. The STM image showing the initial transformation from polymer **1A** to **1B** is presented in Supplementary Fig. 12. The nc-AFM and BR-STM images (Fig. 3$b_{II}$ and 3$b_{III}$) unambiguously reveal the absence of the right-side −CN group at the linkage. The newly-formed C($sp^2$)=C($sp^2$) bond in $L$(C=C) of polymer **1B** is further determined by the bond-length comparison with that of the C($sp^3$)−C($sp^3$) bond in $L$(C−C) of polymer **1 A** (Supplementary Fig. 13). It is important to emphasize that the elimination reaction type in this work is assigned to β-elimination, which involves the dissociation of the −CN group on one carbon atom and the H atom on another carbon atom within the linkage site[33]. Two potential structures derived from α-elimination are discounted due to the absence fingerprints for radical electrons or methylene group in the

experimental nc-AFM characterization results (Supplementary Fig. 14). Moreover, we have discerned significant alterations in the bright contrast of linkages emanating from the partially eliminated intermediate polymer. This observation reinforces the notion that structural planarization is indeed facilitated by the presence of $sp^2$-hybridized carbon atoms (Supplementary Fig. 15). The distance of two adjacent pyrene skeleton in the polymer **1B** is measured to be $d_2 = 23.5 \pm 0.5$ Å (Fig. 3$b_V$), showing good accordance with the DFT-optimized value (23.0 Å) (Fig. 3$b_{VI}$).

In order to obtain C($sp$)≡C($sp$) in $L$(C≡C) via the further elimination of HCN species (step III in Fig. 1a), further annealing to 620 K for 20 min is conducted. The close-up STM image (Fig. 3$c_I$) reveals that the linkage within the polymer **1C** becomes symmetrically linear. Polymer **1C** notates the product of step (III) of **M1**, containing linkages of $L$(C≡C), namely, −C≡C−. The large-scale STM image is exhibited in Supplementary Fig. 16. The distinct bright spots observed in the nc-AFM images (Fig. 3$c_{II}$) offer compelling evidence for the existence of carbon-carbon triple bonds. The BR-STM image (Fig. 3$c_{III}$) exhibits a concave morphology located at the middle of the linkage, which is consistent with the peculiar features of inherent C($sp$)≡C($sp$) bonds in precursor **M1** and is in accordance with the features of the C($sp$)≡C($sp$) bonds characterized by BR-STM in previous studies[24,34]. The Laplace-filtered image matches well with the superimposed chemical structures (Fig. 3$c_{IV}$). The length between two pyrene skeletons within the polymer **1C** is measured to be $d_3 = 23.8 \pm 0.5$ Å (Fig. 3$c_V$), which corresponds well to the DFT-optimized value (23.5 Å) (Fig. 3$c_{VI}$). Both DFT calculated result and experimental observation reveal that the polymers are straightened and elongated along with the stepwise transition of carbon hybridization, which is attributed to variations in bond angles (Fig. 3$a_V$-$c_V$) arising from the three distinct carbon hybridization states (Supplementary Fig. 17). The aforementioned evolution of carbon hybridization is also applicable to the precursors **M2** and **M3** on the same substrate Au(111), as demonstrated in Supplementary Figs. 18 and 19.

Our strategy expands the pathway for constructing COF structures through dehydrogenation reactions[35].

DFT calculations are performed to elucidate the possible reaction pathway on Au(111) utilizing prototype molecules containing −CH₂−CN groups (Fig. 3d, e). Figure 3d, e depict the calculated relevant energies between the reactants, all considered intermediate states, and the final products, respectively. The coordinates of calculational structures are provided in Supplementary Software 1. The reaction pathway of step (I) starts with an initial sub-step where one of the two reactive molecules (labeled as **M M**) dehydrogenates to form the intermediate state **M M•**. The case where one **M** molecule releases two H atoms to form −C:−CN is excluded due to the higher relevant energy. Then two dehydrogenated molecules (labeled as **M•M•**) connect each other to be a dimer with the linkage of $sp^3$-hybridized $L$(C−C). Step (II) involves a direct elimination of −H and −CN groups from each of the reactant molecules in the dimer, which transforms the linkage from $L$(C−C) to $L$(C=C). Another conceivable pathway to releasing −H and −CN groups involves the elimination of two groups at a single reactant molecule (the linkage within this product is denoted as $L$(C−C)*), however, this route is energetically unfavorable. Two other possible pathways for the formation of $L$(C=C) involve eliminating either two −H groups or two −CN groups from the two reactant molecules (the linkage within these products are illustrated as $L$(C=C): and $L$(C=C)**, respectively), but they are excluded due to the energy unfavorability. Finally, step (III) involves another direct elimination of −H and −CN groups, transferring $L$(C=C) to $L$(C≡C). We note that all energy-favored configurations along the proposed reaction pathway (red in Fig. 3d) are observed in experiments.

## The evolution of carbon hybridization across three states on silver surface

To prove the generality of the strategy in multiple-dimensional polymers on different substrates, we perform the experiments on Ag(110) with another precursor **M2** that has a linear molecular skeleton. The schematic illustration is displayed in Fig. 4a. The polymer, being supposed to contain $L$(C−C), is not observed (Supplementary Fig. 20), which may be attributed to the high catalytic activity of Ag(110) and the lower energy barrier for the transformation of linkage from $L$(C−C) to $L$(C=C). Annealing **M2** to 460 K for 20 min on Ag(110) gives rise to the paired polymers with long order (Fig. 4b and Supplementary Fig. 21). High-resolution BR-STM (Fig. 4c) images unveil the inner atomic structures of these polymers which are attributed to polymer **2B** with the linkages $L$(C=C). The polymer **2C**, namely graphyne chain[36], with the uniform $sp$-hybridized linkages $L$(C≡C) is formed after annealing to 520 K, evidenced by STM (Fig. 4d), BR-STM (Fig. 4e), and nc-AFM (Fig. 4f and Supplementary Fig. 22) images.

Notably, polymers **2B** and **2C** embedded with acrylonitrile and ethynylene units can be synthesized with a relatively high selectivity on Ag(110) surface (Supplementary Fig. 21g, h). In contrast to the Knoevenagel reaction[37], we employ solely the −CH₂−CN groups to incorporate the acrylonitrile units into the polymers. Additionally, distinct from the previous work of introducing C($sp$)≡C($sp$) into the nanostructures by utilizing halogen methyl− or halogen alkene−substituted precursors (−CBr₃, −CCl₃, =CBr₂, −CF₃)[10,23,24,38], our work offers a dehydrogenation-initiated, cleaner synthetic route to synthesizing ethynylene-bridged polymers on Ag(110) surface.

Excitingly, the transition of carbon hybridization is also successfully achieved in a 2D COF, utilizing the precursor **M3** on Ag(111) (Fig. 4g and Supplementary Fig. 23). The COF **3B** with the $L$(C=C) linkages are formed via annealing **M3** to 460 K for 20 min. High-resolution STM (Fig. 4h) and BR-STM image (Fig. 4i) reveal the inner structure of COF **3B**. Further annealing to 520 K for 20 min results in the transformation of COF **3B** into COF **3C** with the $L$(C≡C) linkages, as confirmed by STM (Fig. 4j) and BR-STM image (Fig. 4k). The COF **3A** with uniform linkages $L$(C−C) is not detected on Ag(111) but

successfully fabricated on Au(111) (Supplementary Fig. 19). The as-formed COF **3C** exhibits hexagonal units with linear edges of uniform length in its morphology (Fig. 4j, k). However, slight distortion is observed compared to the illustration in Fig. 4g, likely due to the strain retained by the flexible COF, which inhibits its complete relaxation.

To conclude, we successfully realize in-situ visualizations of the atomic configurations and the transformation process of as-formed polymers with distinct carbon linkages in real space by using STM, BR-STM, and nc-AFM imaging measurements, wherein the carbon hybridization evolves across all three states. DFT calculations provide insights into the mechanism underlying the reaction pathways. The electron-withdrawing −CN group plays a unique role in the reaction series by activating C−H bonds of the neighboring saturated methylene and then contributing to the evolution of carbon hybridization through stepwise elimination of HCN species. The realization of regulating carbon hybridization throughout three states in both 1D nonlinear/linear polymers and 2D COFs represents the generality of the strategy developed in this work. The proposed strategy provides an effective approach to regulating carbon hybridization through elaborately designed −CH₂−CN groups, paving the way for the development of new carbon materials for a wide range of applications.

## Methods

### STM and nc-AFM measurements

All STM and nc-AFM images were acquired in a commercial Scienta Omicron low-temperature (4.3 K), ultrahigh vacuum ( $> 1 \times 10^{-10}$ mbar) STM instrument with a tungsten tip placed on a qPlus tuning fork sensor[39]. The CO-functionalized tip was prepared by picking up CO molecule from the previously CO-dosed surface[40]. The bias voltage applied here is the potential of the sample with respect to the tip. The sensor was driven at its resonance frequency (42,500 Hz) with a constant amplitude of 70 pm. The Δz was positive (negative) when the tip-surface distance was increased (decreased) with respect to the STM set point at which the feedback loop was opened. When switching the scanning mode from constant-current to constant-height, ΔZ is set to 0 at that instant. The Au(111) and Ag(111) single crystal surface purchased from MaTeck was prepared in a standard procedure by cycles of Ar⁺ ion sputtering, followed by annealing (750 K < $T$ < 780 K) for 20 min. Prior to the deposition of molecules, the cleanness quality of the pristine gold/silver surfaces were verified with STM. Before the experiment, all the precursors were fully degassed and the sublimation rates of them were determined using a quartz crystal microbalance. Each molecules was sublimated onto the surfaces in situ from quartz crucible in a Knudsen evaporator with 4-fold cells with a required rate. d$I$/d$V$ spectra were acquired with a lock-in amplifier, while the sample bias was modulated by a 699 Hz, 5 mV sinusoidal signal under open-feedback conditions. d$I$/d$V$ maps were acquired in constant-current mode with a CO-tip. Nanotec Electronica WSxM software was used to process images shown here[41].

### Computational details

First-principles calculations based on density functional theory were performed in a plane-wave formulation with the projector augmented wave method, as implemented in the Vienna ab initio simulation package (VASP)[42,43]. Exchange and correlation interactions were treated using the Perdew-Burke-Ernzerhof (PBE) functional[44]. Grimme + 's empirical correction scheme (D3) was also used to take the van der Waals interaction into account[45]. The energy cutoff of the plane-wave basis set was 400 eV. To simulate on-surface adsorption, supercells containing $6 \times 4\sqrt{3}$, three-layer atomic slabs were used to simulate the metal substrates, with molecules adsorbed on one side. Only top-layer Au atoms was shown as wireframes for better visualization. Isosurface: 3.36 × 10⁻³ e/Å³. The vacuum layer is 20 Å between neighboring slabs. The $k$-point sampling was

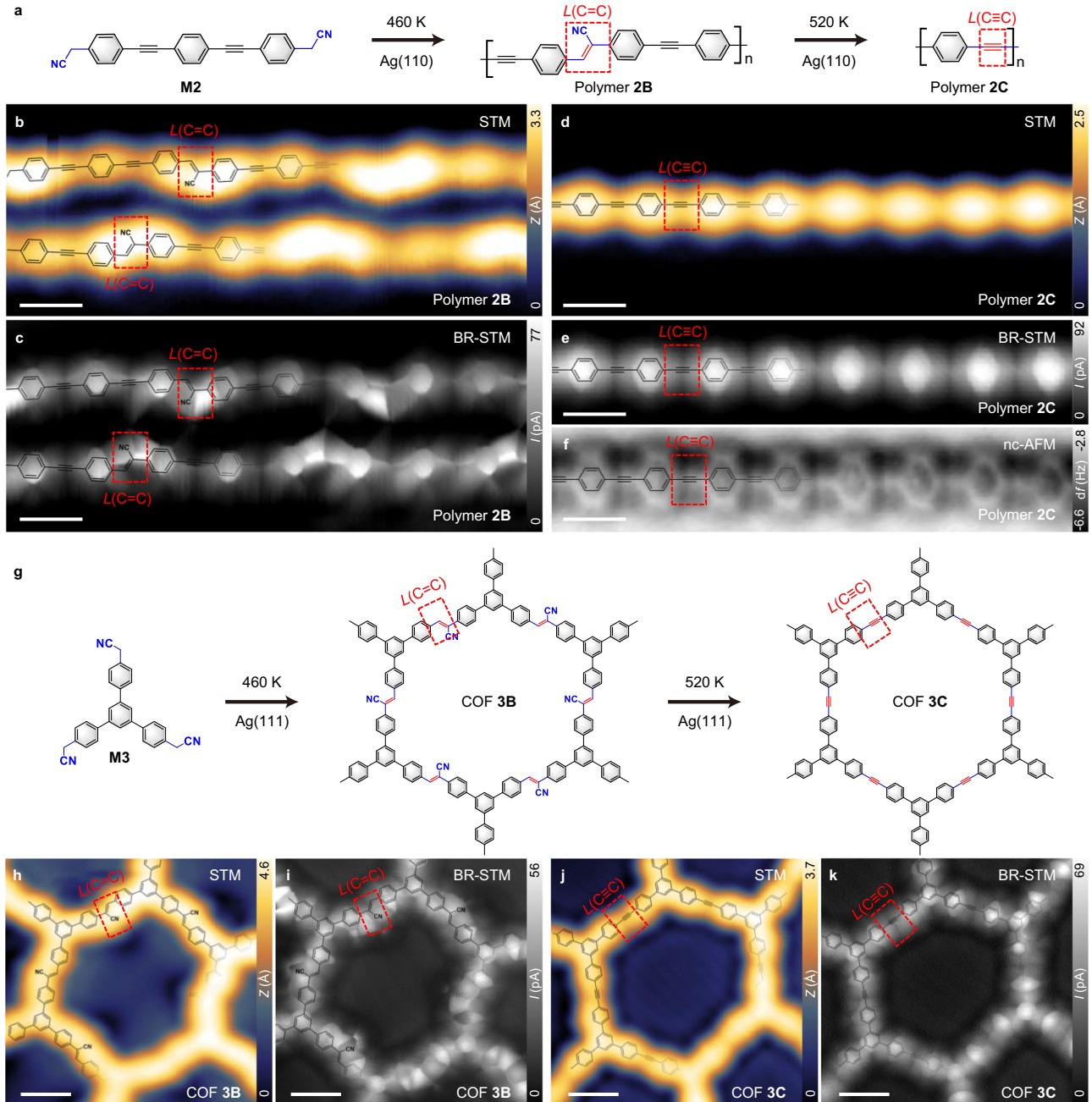

**Fig. 4 | The transition of carbon hybridization within 1D linear polymer and 2D COF fabricated by M2 and M3 on Ag(110) and Ag(111) surfaces, respectively.** **a** Schematic representation showing the synthesis of 1D linear polymers 2B and 2C with the linkages $L$(C=C) and $L$(C≡C) after annealing M2 to 460 K and 520 K for 20 min on Ag(110), respectively. **b, c** High-resolution STM (200 mV, 120 pA) (**b**) and BR-STM (2 mV, Δz = 200 pm) (**c**) images showing the polymers 2B. **d–f** High-resolution STM (200 mV, 120 pA) (d), BR-STM (2 mV, Δz = 160 pm) (**e**), nc-AFM (2 mV, Δz = 190 pm) (f) images showing the polymers 2C obtained. **g** Schematic illustration showing the synthesis of 2D COF 3B and 3C with the linkages $L$(C=C) and $L$(C≡C) after annealing M3 to 460 K and 520 K for 20 min on Ag(111), respectively. **h, i** High-resolution STM (100 mV, 150 pA) (**h**) and BR-STM images (2 mV, Δz = 220 pm) (**i**) showing the COF 3B. **j, k** High-resolution STM (200 mV, 150 pA) (**j**) and BR-STM images (2 mV, Δz = 220 pm) (**k**) showing the COF 3C obtained via annealing to 520 K. Scale bars, 0.6 nm (b–f, h–k).

set Γ-only. In structural relaxations, two bottom layers of metal atoms were fixed, whereas the top-layer metal atoms and the molecules were relaxed until the force on every atom was smaller than 0.01 eV/ Å. Energy barrier calculations for the reactions were performed using the climbing-image-nudged-elastic-band (CI-NEB)[46] methodology as implemented in VASP. The transition state was identified as the configuration with the highest energy along the reaction pathway, where the atomic forces were less than 0.02 eV/Å, and the total energy variation was smaller than $10^{-4}$ eV.

## Data availability
The data that support the findings of this study are available from the corresponding author upon request. The coordinates of calculational structures are provided as Supplementary Software 1. The Source data are provided with this paper.

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

## Acknowledgements

This work was supported by the National Key R&D Program of China (No. 2024YFA1207800); National Natural Science Foundation of China (Nos. 62271238, 22372074, 22105144, 51973153); Yunnan Fundamental Research Projects (No. 202201AT070078); Major Basic Research Project of Science and Technology of Yunnan (202302AG050007); China Postdoctoral Science Fundation (2022M712351); Yunnan Innovation Team of Graphene Mechanism Research and Application Industrializa-tion (202305AS350017); Graphene Application and Engineering

Research Center of Education Department of Yunnan Providence (KKPP202351001); Numerical computations were performed at Hefei Advanced Computing Center.

## Author contributions

J.M.C., J.C.L., L.C. conceived and supervised the experiments; W.X., H.Z., Z.L.H., Z.L.R. designed the on-surface protocols and performed the STM/nc-AFM experiments; G.Z., Y.S.L. synthesized the precursor molecules; D.-L.B., L.G. performed the calculations; J.M.C., J.C.L., L.C., W.X., D.-L.B., and G.Z. analyzed the data; W.X., J.M.C., J.C.L., L.C., H.-J.G., D.-L.B., G.Z. wrote and revised the manuscript.

## Competing interests

The authors declare no competing interests.
