## [Peer Review file · Nature Communications]

Visualizing stepwise evolution of carbon hybridization from sp^3 to sp^2 and to sp

Corresponding Author: Professor Jinming Cai

Version 0:

Reviewer comments:

Reviewer #1

(Remarks to the Author)

The manuscript entitled "Precisely manipulating carbon hybridizations sequentially from sp^3 to sp^2 and to sp " by Wei Xiong et al. reports a study on the on-surface coupling via the HCN elimination on Au(111) and Ag(111) as well as Ag(110) surfaces. The authors stated that this reaction enables the precise manipulation of carbon hybridizations by annealing temperature, which was investigated with a combination of scanning tunneling microscopy, non-contact atomic force microscopy and density functional theory calculations. They showed the pivotal role of the electron-withdrawing cyano ($-CN$) group in activating saturated methylene ($-CH_2-$) groups to form $C(sp^3)-C(sp^3)$ bonds. In general, precise control of carbon hybridization is of crucial interest. However, we (co-review) have significant concerns regarding the data and the presented interpretation. Below, we summarize my comments:

1. The main focus of this work is to control carbon hybridizations through thermal annealing. However, there is no experimental data which proves controllability. At least, the DFT calculation does not seem to support their argument (Figure 3p). It sounds, like other on-surface reactions, that the authors just found the intermediates (single and double) or the final product (triple bond) if annealing at high enough temperature.

2. It is of importance to provide undoubted evidence of bond transformation. The presented images alone do not provide unambiguous characterization of the structures, particularly for polymer 1C and COF 3C. It is highly recommended to conduct NC-AFM imaging and simulations, which can help identify triple bonds as bright dot protrusions, similar to what is shown in Figure 4f.

3. Lines 135-138

The authors stated that dI/dV spectra and dI/dV mapping present analogous electronic states which attest to the homogeneity of the dehydrogenative polymerization. But how we can judge this from the data? To attest to the homogeneity of the dehydrogenative polymerization, the experimental data from NC-AFM and dI/dV cannot exclude other possibilities. Simulation is highly recommended.

4. Lines 147-148

It is true that the dehydrogenation barrier for methylene in phenylacetonitrile is lower than in ethylbenzene. However, 1.44 eV is still high. Are there any factors that lower the barrier? Additionally, how did the authors search for transition states? This information is missing. It appears they simply optimized different structures on the surface and then calculated the energy difference. If this is the case, the barrier cannot be investigated. If not, please provide a more detailed description of how transition states were searched for in the method.

5. Lines 153-154

The authors compared the activation barriers for methylene in phenylacetonitrile and ethylbenzene. It is an interesting finding, but the corresponding experiment data is not shown. It is highly recommended to conduct the reference experiment with ethylbenzene group. Besides this, we also suggest comparing the difference in charge density as well.

6. Line 177

The authors conducted to do the step-by-step annealing. It is highly recommended to include the histograms at each step. The parameters of annealing temperature and time would be also very important to prove the controllability carbon

hybridization through further step-by-step annealing.

7. Line 251

The authors performed experiments on Ag(110) with M2 and on Ag(111) with M3 to demonstrate generality. However, it would be more informative to vary only one parameter at a time. For example, we are curious whether the same reaction can be achieved for M2 and M3 on Au(111) surface.

8. Figure 4

For M3, the COF 3C is apparently not an orthohexagonal structure (structure is distorted), indicating that the linking sites do not seem to be triple bonds. At least, one or two linkers are not triple bonds. It would be nice if the authors could achieve orthohexagonal COF 3C by carefully tuning the annealing temperature. Otherwise, this data does show the less controllability of manipulating carbon hybridization, unfortunately.

9. Figure 2j and Figure 3p

The authors performed DFT calculations to elucidate the possible reaction pathway on Au(111) utilizing prototype molecules containing $-\text{CH}_2-\text{CN}$ groups for both the monomer (Figure 2j) and the dimer (MM in Figure 3p). Why is the dehydrogenation barrier (1.1 eV) of MM lower than that of the monomer (1.44 eV)?

10. Line 283:

In summary, the authors stated that "we successfully manipulate the carbon hybridization across all three states at the linkages". But it is hard to accept this argument. Rather they simply synthesized the three different states by annealing, and the selectivity of $\text{L}(\text{C}=\text{C})$ and $\text{L}(\text{C}\equiv\text{C})$ appears to be relatively low.

Minor comments:

1. In the caption of Figure 2 (lines 101, 102, 105, and 106),

What does " ΔZ " mean? The authors should define where $Z = 0$ is located.

2. In Figure 2e

What is the local regular dip beside the product? Is it induced by CO molecules or the CN group? A discussion on this point is needed.

3. Line 124

A more detailed explanation is needed for why the $-\text{CH}_2-$ groups exhibit local protrusions in the NC-AFM image.

4. Lines 178-180

The authors stated that "STM (Fig. 3a-c) and BR-STM (Fig. 3d-f) characterizations are employed to monitor the transitioning process of carbon hybridization." However, this is not accurate; they cannot monitor the transitioning process and only show the structures of the intermediates.

5. Line 190

The authors stated that the BR-STM image in Figure 3e unambiguously reveals the absence of the right-side $-\text{CN}$ group at the linkage. How did the authors conclude that the contrast around the connection site corresponds to the $-\text{CN}$ group?

Reviewer #2

(Remarks to the Author)

Wei Xiong et al. report in this work a strategy to fabricate polymers taking advantage of tailor-made precursors functionalized with methylcyano ($-\text{CH}_2-\text{CN}$) groups on surfaces. The characterization of the resulting polymers is done by scanning tunneling microscopy and non-contact atomic force microscopy, which helped them to probe the transformation process across three distinct carbon-carbon bonds types: $\text{C}(\text{sp}^3)-\text{C}(\text{sp}^3)$ to $\text{C}(\text{sp}^2)=\text{C}(\text{sp}^2)$ and finally to $\text{C}(\text{sp})\equiv\text{C}(\text{sp})$. I find this work interesting for the general reader of Nature Communications as it illustrates a novel pathway toward the regular formation of carbon-carbon bonds using functional groups unexplored so far on surfaces. The work is technically excellent, however, I consider that there are minor details that should be clarified by the authors before publication.

My main concern is about the way the authors describe the results in the title and abstract. I do not think that it is appropriate or accurate to define it as "precise manipulation". This term is mostly used in experiments where one gets an exquisite control of the chemical changes to be made, i.e. by tip manipulation. In this work it looks like the authors do a standard annealing process with the $\text{C}(\text{sp}^3)-\text{C}(\text{sp}^3)$ and $\text{C}(\text{sp}^2)=\text{C}(\text{sp}^2)$ states as intermediate steps of the final reaction. Therefore I strongly suggest to modify this in the title and related parts of the manuscript. I would actually include the used functional group in the title as I was expecting something completely different after reading it.

My second concern is related to the DFT-calculated energy barriers across the different configurations manuscript. I am not an expert in theoretical calculations, though it was surprising to me that the final state (FS) of all the reactions the authors present are higher in energy than the initial step (IS), which makes no sense to me. Can the authors comment on that?

Reviewer #3

(Remarks to the Author)

Reviewer #4

(Remarks to the Author)

Xiong et al. reports a study on the precise manipulation of carbon hybridizations sequentially transitioning from sp³ to sp² and ultimately to sp states at the linkages within polymers synthesized by several precursors with methylcyano (–CH₂–CN) groups on surfaces. By SPM and DFT measurements, they found that the –CN group, which exhibits electron-withdrawing effect and easy dissociation at higher temperature, is the key factor for triggering dehydrogenation reaction (step I) and elimination reaction (step II/III). They further demonstrated the universal applicability of this strategy on 1D molecular wires as well as 2D COF on Ag(111) and Ag(110) surfaces respectively. The strategy overcomes the constraint that carbon hybridization has previously been confined to transitioning solely between two hybridization types and proposes a new type of on-surface reaction. The manuscript is well written, and the data presented are of high quality. Thus, I recommend its publication in Nature Communications after addressing the following questions:

1. In the description of Figure 2h (line 130-131), the author states “The sp³-hybridized L(C–C) is evidenced by the brighter contrast in the nc-AFM image.” Does this contrast difference in nc-AFM imaging also change after the elimination reaction of the –CN group (step II)? Please provide the corresponding characterization results of the intermediated polymers, if available.
2. As observed in Figure 2e and the supporting information provided by the authors, the M1 precursors undergo not only dehydrogenative polymerization but also crosslinking reactions upon annealing at 520 K on Au(111) surface. Can the authors explain the reasons behind this crosslinking reaction? And are there any other strategies that can be employed to reduce the yield of such crosslinking structures?
3. The elimination reaction induced by the dissociation of –CN group is crucial for the transformation of connection sites within polymers. The authors need to clarify the type of elimination reaction in this work based on previous reports about on-surface elimination reactions [J. Am. Chem. Soc. 2023, 145, 11, 6203–6209].
4. The authors stated that 1D and 2D polymers containing L(C–C) linkages cannot be observed on the Ag(110) and Ag(111) surfaces, but the currently presented experimental results were not convincing enough to support this conclusion. Can the authors provide more detailed experimental results at lower temperatures that initially triggered the reaction on Ag(111) or Ag(110) surfaces?
5. Can the sequential transition of carbon hybridization also be achieved within COF structures on Au(111) surface?
6. Based on previous reports [ACS Nano 2017, 11, 7355–7361], –CN groups tend to undergo cyclization reactions with aromatic structures during the annealing process. Why does the cyano group exhibit a preference for dissociating from the linkage site and undergoing an elimination reaction, rather than engaging in a cyclization reaction with the adjacent phenyl group?
7. Regarding reaction barriers, typically the product shall have less free energy. However, in the calculated energy path the final product has a relatively higher energy than the others in the path. Please give some comments.
8. In calculations, are those small prototype molecules able to reasonably describe the energy barriers for the realistic big molecules?
9. Some minors: (1) The full name of BR-STM is needed; (2) Two Supplementary Figs. 13.

Reviewer #5

(Remarks to the Author)

Version 1:

Reviewer comments:

Reviewer #1

(Remarks to the Author)

We (co-reviewers) appreciate the efforts that the authors addressed to our concerns in the initial manuscript. It is obvious that simulated dI/dV mappings based on the DFT calculations and the precise experiment on the DBP molecule and new precursor M4 improved the manuscript significantly. The authors also conducted a non-contact atomic force microscopy (nc-AFM) experiment on polymer 1C, providing compelling evidence of the bond transformation. However, they mention that achieving an orthohexagonal COF 3C structure is challenging. If this is the case, the schematic illustration of COF 3C in Figure 4g and the corresponding description may no longer be appropriate. We think that this point should be clarified in the manuscript before accepting this contribution for the publication.

Reviewer #2

(Remarks to the Author)

The authors have successfully replied to all my comments and, in my opinion, the manuscript is ready to be published in Nature Communications.

Reviewer #3

(Remarks to the Author)

Reviewer #4

(Remarks to the Author)

The authors exerted significant effort in revising the manuscript. All my concerns and the others have been well addressed, and the quality of this revised manuscript has been improved. I thus recommend it for publication in Nature Communications.

Reviewer #5

(Remarks to the Author)

A point-by-point response to the reviewers' comments

We thank all reviewers for the positive and constructive comments and suggestions. We have addressed all the comments point-by-point and revised the manuscript accordingly. In this response letter, comments from the reviewers are summarized in blue typeface. Our responses are in normal black font. The text in the manuscript and supporting information are presented in *italics* and the revised parts are highlighted with yellow shadow.

Response to reviewer #1

The manuscript entitled “Precisely manipulating carbon hybridizations sequentially from sp^3 to sp^2 and to sp ” by Wei Xiong et al. reports a study on the on-surface coupling via the HCN elimination on Au(111) and Ag(111) as well as Ag(110) surfaces. The authors stated that this reaction enables the precise manipulation of carbon hybridizations by annealing temperature, which was investigated with a combination of scanning tunneling microscopy, non-contact atomic force microscopy and density functional theory calculations. They showed the pivotal role of the electron-withdrawing cyano ($-CN$) group in activating saturated methylene ($-CH_2-$) groups to form $C(sp^3)-C(sp^3)$ bonds. In general, precise control of carbon hybridization is of crucial interest. However, we (co-review) have significant concerns regarding the data and the presented interpretation. Below, we summarize my comments:

Comment 1

The main focus of this work is to control carbon hybridizations through thermal annealing. However, there is no experimental data which proves controllability. At least, the DFT calculation does not seem to support their argument (Figure 3p). It sounds, like other on-surface reactions, that the authors just found the intermediates (single and double) or the final product (triple bond) if annealing at high enough temperature.

Author reply:

We appreciate this suggestion from the reviewer. We have thoroughly reviewed our results and agree with the reviewer's assessment that the claims of “precise manipulation” and “high controllability” exceed the scope of our findings. Accordingly, we have toned down the related claims, including rephrasing the title, introduction, and related discussions in the main text. The relevant revisions are as follows:

The title is revised to be “*Visualizing stepwise evolution of carbon hybridization from sp^3 to*”

sp² and to sp”

Modifications to the abstract:

“~~Precise manipulation of carbon’s three~~ Regulating three carbon hybridization states lies at the heart of engineering carbon materials with tailored properties^{1,2}.”

“..., ~~meticulously~~, orchestrating the sequential transition across ...”

“Surface science provides an exceptional platform for ~~the precise manipulation and visualization of controlling and visualizing~~ carbon hybridization at the atomic level by leveraging scanning probe microscopy^{7,10,11}.”

“Here, we ~~report a strategy to precisely manipulating carbon hybridizations in a controlled, stepwise manner, sequentially transitioning~~ present the visualization of stepwise evolution in carbon hybridizations, from sp³ to sp² and ultimately to sp states...”

“..., we unequivocally probe the ~~thermally induced~~ transformation process across three distinct carbon-carbon bond types:...”

“~~Thermal annealing enables precise control over the entire reaction, conducting high yields of each bond type at corresponding steps.~~”

“Our work pioneers a breakthrough ... and serves as a dramatic advance ..., ~~enabling unprecedented controllability and flexibility~~ by employing novel cyanomethyl-substituted organic compounds.”

Modifications to the main text:

In several instances, the wording “manipulating” and “manipulation” has been replaced with “regulating” and “regulation”.

Line 57 *“Fig. 1 | Illustration of sequentially ~~manipulating~~ regulating carbon ~~bond~~ hybridization states ~~by employing of~~ the (4-methylcyano)phenyl-substituted compounds on surface.”*

Line 67 *“The first and pivotal step towards ~~manipulating~~ regulating...”*

Line 90 *“(i) The -CH₂-CN groups within **M1**, **M2** and **M3** allow the formation and ~~manipulation~~ regulation of carbon-carbon bonds...”*

Line 307 *“Notably, polymers **2B** and **2C** embedded with acrylonitrile and ethynylene units can be ~~selectively~~ synthesized with a ~~relatively high yield~~ selectivity on Ag(110) surface (Extended*

Data Fig. 5g-h).”

Line 311 “Additionally, distinct from..., our work offers a dehydrogenation-initiated, cleaner synthetic route to synthesizing ethynylene-bridged polymers ~~with exceptional quality and length (submicron level)~~ on Ag(110) surface.”

Modifications to the Conclusion:

“To conclude, we successfully ~~manipulate the carbon hybridization across all three states at the linkages within polymers fabricated by precursors containing CH₂-CN groups on surfaces. By using STM, BR-STM, and nc-AFM imaging measurements, we realize in-situ visualizations of the atomic configurations and the transformation process of as-formed polymers with distinct carbon linkages in real space~~ by using STM, BR-STM, and nc-AFM imaging measurements, wherein the carbon hybridization evolves across all three states.”

“~~The characterizations, complemented by DFT calculations,~~ provide insights into the mechanism underlying the reaction pathways.”

“The electron-withdrawing -CN group... contributing to the ~~manipulation~~ evolution of...”

“The realization of ~~precisely manipulating~~ regulating carbon hybridization...”

“The proposed strategy represents a pioneering approach to ~~regulating~~ ~~manipulating~~ carbon hybridization...”

Comment 2

It is of importance to provide undoubted evidence of bond transformation. The presented images alone do not provide unambiguous characterization of the structures, particularly for polymer 1C and COF 3C. It is highly recommended to conduct NC-AFM imaging and simulations, which can help identify triple bonds as bright dot protrusions, similar to what is shown in Figure 4f.

Author reply:

We appreciate the reviewer’s attention to the importance of undoubted evidence for bond transformation. We have performed nc-AFM characterizations and simulations on polymer 1C, which confirm the identification of triple bonds as bright dot protrusions (Fig. R1).

Fig. R1 | **a-b**, Experimental (**a**) and simulated (**b**) nc-AFM images of polymer **1C**. Scanning parameters: 2 mV, $\Delta z = 200$ pm. We note that experimental measurements agree with theoretical simulations, both identifying triple bonds as bright dot protrusions. Simulation setup: Simulations of nc-AFM were performed with a modification of the particle probe model code by Hapala et al [*Phys. Rev. Lett.* 119, 166001 (2017)] The structure and electrostatic potential were calculated with DFT. The tip was modeled with a CO molecule with a charge of $-0.05 e$, suspending upon the molecule by 3.8 \AA height. A dz^2 quadrupole model was considered. The tip lateral spring constant is 0.2 N/m .

We have also integrated nc-AFM images of polymers **1A**, **1B** and **1C** as Fig. 3(a_{II}-c_{II}) in the manuscript and revised the figure caption, accordingly.

Fig. 3 | **Stepwise transition of carbon hybridization across all three states at the linkage via annealing to elevated temperatures. a-e**, STM images (50 mV, 200 pA) showing the polymers **1A**, **1B**, **1C** with the distinct linkages $L(C-C)$, $L(C=C)$, and $L(C\equiv C)$ via annealing to 520 K,

550 K, and 620 K, respectively. **d-f**, BR-STM images (2 mV, $\Delta z = 160$ pm) corresponding to (**a-e**). **g-i**, Laplace-filtered images of polymers corresponding to (**d-f**), which are superimposed by chemical structures. **j-l**, Laplace-filtered images of polymers corresponding to (**a-e**), wherein the bond angles of carbon-carbon bonds and spacings between adjacent pyrene skeletons are measured. **m-o**, DFT-optimized structures of polymers **1A**, **1B**, and **1C**. **a_I-a_{VI}**, Polymer **1A** with the linkage L(C-C). **b_I-b_{VI}**, Polymer **1B** with the linkage L(C=C). **c_I-c_{VI}**, Polymer **1C** with the linkage L(C≡C). STM images (**a_I-c_I**), nc-AFM images (**a_{II}-c_{II}**), BR-STM images (**a_{III}-c_{III}**), Laplace-filtered BR-STM images overlaid with chemical structures (**a_{IV}-c_{IV}**), Laplace-filtered BR-STM images (**a_V-c_V**) showing distinct bond angles and relative distances in three polymers, and DFT-optimized structures (**a_{VI}-c_{VI}**). Scanning parameters: (**a_I-c_I**) 200 mV, 200 pA. (**a_{II}-c_{II}**) 2 mV, $\Delta z = 210$ pm. (**a_{III}-c_{III}**) 2 mV, $\Delta z = 160$ pm. **p, d**, DFT-calculated relative energies for corresponding configurations. The solid line presents the most possible reacting pathway, in which the species highlighted in red are observed in experiments. While dashed lines present other considered paths. **q, e**, Illustrations of the configurations in (**p, d**). Scale bar, 0.4 nm (**a_I-c_V**).

We are unable to provide the corresponding nc-AFM image of COF **3C**, owing to the challenges encountered in obtaining an orthohexagonal COF **3C** structure.

Finally, we have appended some descriptions about the nc-AFM characterizations for polymers **1A**, **1B**, and **1C**, as outlined below:

Line 204 “STM (Fig. 3 **a-e a_I-c_I**), nc-AFM (Fig. 3 **a_{II}-c_{II}**) and BR-STM (Fig. 3 **d-f a_{III}-c_{III}**) characterizations are employed to ~~monitor the transitioning process of carbon hybridization~~ characterize the corresponding products.”

Line 217 “The nc-AFM and BR-STM images (Fig. 3 **e b_{II}** and **3b_{III}**) unambiguously reveals the absence of the right-side -CN group at the linkage.”

Line 241 “The distinct bright spots observed in the nc-AFM images offer compelling evidence for the existence of carbon-carbon triple bonds.”

Comment 3

Lines 135-138: The authors stated that dI/dV spectra and dI/dV mapping present analogous electronic states which attest to the homogeneity of the dehydrogenative polymerization. But how we can judge this from the data? To attest to the homogeneity of the dehydrogenative polymerization, the experimental data from NC-AFM and dI/dV cannot exclude other possibilities. Simulation is highly recommended.

Author reply:

To better illustrate the homogeneity of as-formed dehydrogenative polymerization, we performed detailed DFT calculations on STM simulations of polymer **1A**, which as $L(C-C)$ at linkages, with corresponding bias voltage. We have added new results to the Extended Figure 1 as panels e, f, and g. The updated Extended Figure 1 is as the follows.

Extended Data Fig. 1 | Electronic structures of the polymer 1A on Au(111). *a*, dI/dV spectra recorded on two adjacent linkages of the polymer **1A**. The red and blue curves are from the corresponding colored spots in (**b**). The grey curve is taken on a bare Au(111) surface. **b**, STM image (400 mV, 20 pA) of the polymer **1A** superimposed by the corresponding chemical structures. The blue and pink dots represent the positions of spectra acquisition. **c-d**, Constant-current dI/dV maps conducted at potential energies of -1100 mV (**c**) and 1100 mV (**d**), respectively. dI/dV maps are superimposed by chemical structures to enhance the visualization of frontier molecular orbitals distribution. Scale bars, 1 nm (**b-d**). **e**, Top view of the atomic model of freestanding polymer **1A** chain. **f-g**, Simulated dI/dV mappings of polymer **1A**.

We highlight the similarity between the experiment and the simulation, particularly the bright protrusions at the $-CN$ functional groups, which are upwards from the substrate. Notably, the electron density around $-CN$ couples with the adjacent phenyl rings, forming an arc-like morphology on both sides, a feature also captured in simulations (dashed green arrows in Extended Data Figs. 1c and f). DFT-calculated dI/dV mapping simulation acquired at negative bias of -2 V present that two upward-going $-CN$ functional groups appear as isolated bright dots, and the pyrene group show comparable brightness, consistent with the experimental dI/dV mapping in (**d**). It is important to note that due to the doping effect of the substrate and future planeness of molecules on substrate, the bias voltages in simulations are not quantitatively comparable to those in the experiments.

We made considerable efforts to perform nc-AFM simulations of polymer **1A**; however, due to

the upward orientation of the –CN groups, the simulations are non-trivial and were unable to provide clear evidence to verify the structural information.

Comment 4

Lines 147-148: It is true that the dehydrogenation barrier for methylene in phenylmethylcyano is lower than in ethylbenzene. However, 1.44 eV is still high. Are there any factors that lower the barrier? Additionally, how did the authors search for transition states? This information is missing. It appears they simply optimized different structures on the surface and then calculated the energy difference. If this is the case, the barrier cannot be investigated. If not, please provide a more detailed description of how transition states were searched for in the method.

Author reply:

We thank the reviewer for his careful suggestions. We agree that 1.44 eV is still a relatively high barrier for real reactions. We would like to note, though, the focus of calculated dehydrogenation barriers between the two prototype molecules is to illustrate the electron-withdrawing nature of –CN, not to calculate out realistic energy barriers for the experimentally investigated molecules. Regard the specific calculated values, reaction pathways with calculated energy barriers above 1 eV, specifically, 1.33 eV [*Angew. Chem. Int. Ed.* 61, e202212354 (2022).], 1.80 eV [*Nat. Commun.* 15, 1910 (2024).], 1.60 eV [*Nat. Synth.* 1, 289–296 (2022).], and so on, have been widely accepted to underlie the mechanism of experimentally observed on-surface reactions. Experiments are more complicated than simulation modelling. There are a lot of possible factors that has been reported to be able to further lower the barriers of on-surface reactions, like metal adatoms [*Angew. Chem. Int. Ed.* 61, e202212354 (2022).], parading H atoms [*Nano Res.* 14, 4563–4568 (2021)], local electric field around the STM tip [*J. Am. Chem. Soc.* 129, 4298–4305 (2007)] and so on.

Regard the method used to search for transition states, we have added the following to the method section.

Line 379 “Energy barrier calculations for the reactions were performed using the climbing-nudged-elastic-band (CI-NEB)³⁴ methodology as implemented in VASP. The transition state was identified as the configuration with the highest energy along the reaction pathway, where the atomic forces were less than 0.02 eV/Å, and the total energy variation was smaller than 10⁻⁴ eV.”

Comment 5

Lines 153-154: The authors compared the activation barriers for methylene in phenylmethylcyano and ethylbenzene. It is an interesting finding, but the corresponding experiment data is not shown. It is highly recommended to conduct the reference experiment with ethylbenzene group. Besides this, we also suggest comparing the difference in charge density as well.

Author reply:

We agree with suggestions of performing control experiments and calculations to further support our conclusions. We have designed the 4,4'-diethylbiphenyl (**DBP**) molecule for comparison with methylcyano-functionalized molecules in terms of their selectivity for methylene dehydrogenative homocoupling on Au(111) surface. The corresponding results have been incorporated into the supplementary information, as illustrated in Supplementary Fig. 8.

Supplementary Fig. 8 | A control experiment of 4,4'-diethylbiphenyl (DBP) molecule on Au(111) substrate. a, Chemical structure of DBP molecule. b, STM image showing the self-assembled DBP molecules on Au(111) surface (200 mV, 120 pA). c-d, Overview and magnified STM images (200 mV, 50 pA) showing the edge-fused by-products after annealing DBP molecules to 520 K.

Supplementary Fig. 8a present the chemical structure of **DBP** molecule functionalized with two $-\text{CH}_2-\text{CH}_3$ groups. Initially, upon the deposition of **DBP** molecules onto Au(111) surface

held at room temperature, island-like self-assembly structures are formed (as shown in Supplementary Fig. 8b). Subsequently, annealing the sample to 520 K triggers polymerization reactions among some **DBP** molecules (Supplementary Fig. 8c). From the magnified image, it can be observed that the regions marked by white dashed ellipses are all edge-fused products, indicating heterogeneous reaction sites, i.e., the terminal methyl groups cannot activate adjacent methylene groups (Supplementary Fig. 8d). Moreover, Chi *et.al.* also did not observe the dehydrogenative homocoupling between methylene groups during the synthesis of hexabenzocoronene-cored graphdiyne nanosheets using hexa(4-ethylphenyl)benzene (**HPB-Et**) molecules [*Angew. Chem. Int. Ed.* 63, e202411722 (2024)]. Based on the comparative experimental results between **M1** and **DBP** molecules, we believe that the cyano group plays a crucial role in activating the methylene group.

We have added relevant descriptions to the manuscript as follows:

Line 162 “*Additionally, we specifically designed the molecule 4,4'-diethylbiphenyl (DBP) to conduct a comparative analysis of the selectivity displayed in the dehydrogenation reactions of $-CH_2-$ groups on Au(111) surface, as opposed to methylcyano-functionalized molecules. Our experimental results indicate that, unlike $-CN$ groups, the terminal $-CH_3$ groups are unable to selectively activate adjacent $-CH_2-$ groups (see Supplementary Fig. 8 for further details).*”

We have added the following analysis to the charge density differences of two prototype molecules on Au(111) substrate to the supporting information, shown as Supplementary Fig. 10.

Supplementary Fig. 10 | DFT-calculated charge density differences of phenylacetonitrile and ethylbenzene on Au(111) surface. a, Schematic of phenylacetonitrile on Au(111). b-c, Top (b)

and side (c) views of charge density difference between phenylacetonitrile molecule and substrate, like the main text Fig. 2j. **d**, Schematic of ethylbenzene on Au(111). **e-f**, Top (e) and side (f) views of charge density difference between ethylbenzene molecule and substrate. In (b-c) and (e-f), purple isosurface presents electron depletion, while green isosurface presents electron accumulation. Isosurface value: $6.7 \times 10^{-3} e/\text{\AA}^3$.

We emphasize that the charge density distributions for the two molecules exhibit distinct differences. As discussed in the main text, for phenylacetonitrile, there is a region of charge depletion between the $-\text{CN}$ group and the substrate due to the electron-withdrawing nature of the $-\text{CN}$ functional group (highlighted by blue dashed curves in panels (b-c)). This leads to charge accumulation between the $-\text{CH}_2-$ group and the substrate, which facilitates dehydrogenation (highlighted by black dashed curves in panels (b-c)).

In contrast, for ethylbenzene on Au(111) surface, there is no significant charge redistribution between the functional groups and the substrate, indicating no facilitation of dehydrogenation (highlighted by pink and black curves in panels (e-f)).

We have supplemented the relevant descriptions in the manuscript, as shown below:

Line 178 “*Conversely, in the case of ethylbenzene on Au(111), there is an insignificant charge redistribution observed between the $-\text{CH}_2-\text{CN}$ groups and the substrate, suggesting no enhancement in the dehydrogenation process (Supplementary Fig. 10).*”

Comment 6

Line 177: The authors conducted to do the step-by-step annealing. It is highly recommended to include the histograms at each step. The parameters of annealing temperature and time would be also very important to prove the controllability carbon hybridization through further step-by-step annealing.

Author reply:

We believe the histograms mentioned by the reviewer is referring to the product yield under different annealing temperatures. Therefore, we have conducted a statistical analysis of the yields of molecules of **M1** and **M2** at varying temperatures on Au(111) surface to elucidate the effectiveness of annealing strategy.

The following is results of polymers **1A**, **1B** and **1C** obtained from **M1** molecules at different temperatures, as shown in Fig. R2.

Fig. R2 | **a-c**, Large-scale STM images (200 mV, 50 pA) showing the samples on Au(111) surface obtained *via* annealing **M1** to 520 K, 550 K and 620 K for 20 mins, respectively. **d**, Yield distribution statistics of polymer **1A**, **1B** and **1C** obtained *via* annealing to 520 K, 550 K and 620 K for 20 mins, respectively.

It can be observed that the yield of polymer **1A** arising from the first dehydrogenative polymerization step (520 K, 20 minutes) is dominantly high (~93%). As the temperature increases to 550 K, polymer **1B** turns to be the majority product with a yield of ~60 % (Annealing time: 20 minutes). When the samples are annealed at 620 K for 20 minutes, polymer **1C** is the majority product with a yield of ~55 %. We note that more deformed structures are formed as the temperature increases. It is likely the consequence of **M1** molecules dissociation, which is molecule- and substrate-dependent.

To more broadly characterize the relationship between carbon hybridization evolution rates and annealing steps, we performed systematic analyses using additional molecules, **M2**, on Au(111) substrate, and added related results and discussion to the supplementary information as Supplementary Figure 14.

Supplementary Fig. 14 | Annealing M2 to different temperatures on Au(111) surface. a, Schematic representation showing the synthesis of polymers **2A**, **2B** and **2C**. **b-e,** Overview STM images (200 mV, 120 pA) showing the self-assembled **M2** molecules (**b**), polymer **2A** (**c**), **2B** (**d**) and **2C** (**e**) on Au(111) surface after annealing the sample to RT, 520 K, 550 K and 620 K for 20 mins, respectively. **f-h,** Magnified STM images (200 mV, 250 pA) showing the structures corresponding to (**b-e**) (200 mV, 250 pA). **i-j,** STM (200 mV, 250 pA) and BR-STM image (2 mV, $\Delta z = 200$ pm) of polymer **2C**. **k,** Yield statistics for the products resulting from the dehydrogenation reaction of **M2** molecules after annealing at 500 K and 520 K for 20 mins. **l,** Yield statistics for the products obtained from the two-step elimination reactions of **M2** molecules after annealing to 550 K and 620 K for 20 mins.

The yields of polymers **2A**, **2B**, and **2C** on the Au(111) surface after annealing to 520 K, 550 K, and 620 K, respectively, were significantly higher compared to the yields of **M1** on Au(111) surface. This demonstrates the effectiveness of the stepwise annealing strategy. The reaction

rates could be controlled through both molecular structure and substrate selection.

Then we would like to emphasize that the annealing temperature has no substantial impact on the selectivity of the reaction. In the main text, we have chosen 20 minutes as the holding time for the preparation of several polymers. The detailed annealing parameters and time into the manuscript, as follows:

Line 105 “*e*, Large-scale STM...after annealing to 520 K for 20 mins.”

Line 131 “The reaction is triggered upon annealing the self-assembled structure to 520 K for 20 mins.”

Line 211 “Upon annealing to 550 K for 20 mins, it is...”

Line 236 “In order to ..., further annealing to 620 K for 20 mins is conducted.”

Line 282 “*a*, Schematic representation...after annealing **M2** to 460 K and 520 K for 20 mins on Ag(110), respectively.”

Line 288 “*g*, Schematic illustration showing...after annealing **M3** to 460 K and 520 K for 20 mins on Ag(111), respectively.”

Line 300 “Annealing **M2** to 460 K for 20 mins on Ag(110) gives rise to the paired polymers with long order (Fig. 4b and Extended Data Fig. 5).”

Line 317 “The COF **3B** with the L(C=C) linkages are formed via annealing **M3** to 460 K for 20 mins.”

Line 320 “Further annealing to 520 K for 20 mins results in...”

Comment 7

Line 251: The authors performed experiments on Ag(110) with **M2** and on Ag(111) with **M3** to demonstrate generality. However, it would be more informative to vary only one parameter at a time. For example, we are curious whether the same reaction can be achieved for **M2** and **M3** on Au(111) surface.

Author reply:

We appreciate this suggestion from the reviewer. We have performed the experiments on stepwise evolution of carbon hybridization of 1D and 2D polymers synthesized by **M2** and **M3** molecules on Au(111) surface via annealing to elevated temperatures, as shown in Supplementary Figs. 14 and 15.

Although Supplementary Figure 14 was previously provided in response to the earlier comment, we include it here again for your ease of reference.

Supplementary Fig. 14 | Annealing M2 to different temperatures on Au(111) surface. a, Schematic representation showing the synthesis of polymers **2A**, **2B** and **2C**. **b-e**, Overview STM images (200 mV, 120 pA) showing the self-assembled **M2** molecules (**b**), polymer **2A** (**c**), **2B** (**d**) and **2C** (**e**) on Au(111) surface after annealing the sample to RT, 520 K, 550 K and 620 K for 20 mins, respectively. **f-h**, Magnified STM images (200 mV, 250 pA) showing the structures corresponding to (**b-e**) (200 mV, 250 pA). **i-j**, STM (200 mV, 250 pA) and BR-STM image (2 mV, $\Delta z = 200$ pm) of polymer **2C**. **k**, Yield statistics for the products resulting from the dehydrogenation reaction of **M2** molecules after annealing at 500 K and 520 K for 20 mins. **l**, Yield statistics for the products obtained from the two-step elimination reactions of **M2** molecules after annealing to 550 K and 620 K for 20 mins.

Supplementary Fig. 15 | *The transition of carbon hybridization within 2D COFs fabricated via annealing M3 to different temperatures on Au(111). a-c, Large-scale STM images (200 mV, 150 pA) showing the COF structures fabricated by annealing M3 to 520 K (a), 550 K (b), and 620 K (c) for 20 mins on Au(111), respectively. d-f, Magnified STM image (400 mV, 150 pA) derived from (a-c).*

It has been demonstrated that dehydrogenative polymerization and stepwise carbon hybridization evolution can also be achieved with **M2** and **M3** on Au(111). The one-dimensional nature of polymerized **M2** enhances spatial confinement, producing a more regular and analyzable linear molecular chain. In contrast, **M3** on Au(111) yields less controllable products, as the two-dimensional COF leads to more complex arrangements, providing more sites for by-products. Additionally, the weaker catalytic performance of Au(111) results in lower-quality polymerized **M3** compared to those on Ag(111) as reported in the main text.

We have added relevant descriptions into the manuscript as follows:

Line 253 “*The aforementioned evolution of carbon hybridization is also applicable to the precursors M2 and M3 on the same substrate Au(111), as demonstrated in Supplementary Figs. 14 and 15. To the best of our knowledge, there have been no previous reports of COF structures*

containing succinonitrile units constructed through dehydrogenation reactions, either in on-surface synthesis or in solution-phase chemistry²³.”

Line 322 “*The COF 3A ...but successfully fabricated on Au(111) (Supplementary Fig. 4015).*”

Line 323 “*It is worth emphasizing that there is a lack of reports regarding the construction of COF structures through dehydrogenation reactions in both on-surface and wet chemistry as far as we know²².*”

Comment 8

Figure 4

For M3, the COF 3C is apparently not an orthohexagonal structure (structure is distorted), indicating that the linking sites do not seem to be triple bonds. At least, one or two linkers are not triple bonds. It would be nice if the authors could achieve orthohexagonal COF 3C by carefully tuning the annealing temperature. Otherwise, this data does show the less controllability of manipulating carbon hybridization, unfortunately.

Author reply:

We are grateful to the reviewer for bringing this matter to our attention. We conducted additional, more precise experiments in an effort to obtain a higher yield of orthohexagonal COF; however, the results did not improve compared to those presented in Fig. 4 in the manuscript.

We would like to note that the carbon hybridization can be elucidated through morphology changes. The transformation of edges from curved to linear forms clearly demonstrates bonding changes, as illustrated in the main text Figs. 4h-k.

We attribute the distortion of as-formed hexagonal framework to strain. While we expect that annealing could initiate linkage formation and stepwise evolution at chemically active sites, it appears insufficient to relieve the large-scale strain within the COF. This is particularly the case given that the COF forms from relatively random molecular assembly and the edges of the framework are flexible single chains.

To further elucidate how strain effects the imaging morphology of $-C\equiv C-$ bonds, we designed a new precursor **M4**, which inherently contains triple bonds, and performed the polymerization experiments on Ag(111), as shown in Fig. R3.

Fig. R3 | On-surface synthesis of polymer 4C with the utilization of M4 molecules on Ag(111) surface. **a**, Schematic illustration of synthesizing polymer 4C on Ag(111) bridged by $L(C\equiv C)$ linkages in utilization of M4 precursors. **b**, STM image showing the polymer 4C on Ag(111) surface (200 mV, 120 pA). **c-e**, STM image (200 mV, 100 pA) (**c**), BR-STM image (2 mV, $\Delta z = 170$ pm) (**d**) and Laplace-filtered image (**e**) of polymer 4C.

It is evident that the intrinsic carbon-carbon triple bonds within the polymer 4C (as marked by red arrows) are distorted because of the influence of the deformed upper ending, which is attributed to the inherent flexibility of carbon-carbon triple bonds [*Angew. Chem. Int. Ed.* 2019, 58, 6559]. Therefore, some linkages presenting bending characteristic in the BR-STM image of COF 3C shown in Fig. 3k of the manuscript may also be assigned to carbon-carbon triple bonds.

Comment 9

Figure 2j and Figure 3p The authors performed DFT calculations to elucidate the possible reaction pathway on Au(111) utilizing prototype molecules containing $-CH_2-CN$ groups for both the monomer (Figure 2j) and the dimer (MM in Figure 3p). Why is the dehydrogenation barrier (1.1 eV) of MM lower than that of the monomer (1.44 eV)?

Author reply:

We thank the reviewers for providing constructive feedback and we would like to clarify that

Fig. 3q illustrates the energy profile along possible reaction paths, not the energy barriers. The 1.1 eV value represents the energy difference between the two configurations of **MM** and **MM•**. A larger energy barrier, exceeding 1.1 eV, is expected for the transition from **MM** to **MM•**. We propose that the likely reaction path for this step involves one **M** molecule undergoing dehydrogenation to form **M•**, with an energy cost of 1.44 eV as calculated in Figure 2j. The dehydrogenated **M•** then attracts another **M** molecule, moving closer and then forming the next intermediate state **M•M•**. It's important to note that the actual dehydrogenation barrier is likely reduced by external factors, as outlined in our response to Reviewer 1, Comment 4. These factors may include metal adatoms [*Angew. Chem. Int. Ed.* 63, e202411722 (2024).], parading H atoms [*Nano Res.* 14, 4563-4568 (2021).], or the local electric field around the STM tip [*J. Am. Chem. Soc.* 129, 4298-4305 (2007).], rather than the configuration of dimer.

Comment 10

Line 283: In summary, the authors stated that “we successfully manipulate the carbon hybridization across all three states at the linkages”. But it is hard to accept this argument. Rather they simply synthesized the three different states by annealing, and the selectivity of $L(C=C)$ and $L(C\equiv C)$ appears to be relatively low.

Author reply:

We thank the reviewer to bring this to our attention again. Referencing to our response to the Comment 1, we have toned down the claim about “precise controllability” and made corresponding revisions through the main text, including the content in the conclusion section. Here we copy the revision again for reviewer’s convenience of reference.

“To conclude, we successfully ~~manipulate the carbon hybridization across all three states at the linkages within polymers fabricated by precursors containing $-CH_2-$ CN groups on surfaces. By using STM, BR-STM, and nc-AFM imaging measurements, we realize in-situ visualizations of the atomic configurations and the transformation process of as-formed polymers with distinct carbon linkages in real space by using STM, BR-STM, and nc-AFM imaging measurements, wherein the carbon hybridization evolves across all three states.”~~

“~~The characterizations, complemented by DFT calculations,~~ provide insights into the mechanism underlying the reaction pathways.”

“The electron-withdrawing $-CN$ group... contributing to the ~~manipulation~~ evolution of...”

“The realization of ~~precisely manipulating~~ regulating carbon hybridization...”

“The proposed strategy represents a pioneering approach to regulating manipulating carbon hybridization...”

Comment 11

In the caption of Figure 2 (lines 101, 102, 105, and 106), What does “ ΔZ ” mean? The authors should define where $Z = 0$ is located.

Author reply:

ΔZ represents the distance between the tip and the sample in the constant-height mode. When switching the scanning mode from constant-current to constant-height, ΔZ is set to 0 at that instant. Subsequently, adjustments to the ΔZ value are made based on the image quality obtained from BR-STM and nc-AFM. We have supplemented relevant information in the Method section as follows:

Line 353 “When switching the scanning mode from constant-current to constant-height, ΔZ is set to 0 at that instant.”

Comment 12

In Figure 2e, What is the local regular dip beside the product? Is it induced by CO molecules or the CN group? A discussion on this point is needed.

Author reply:

The depressions that are regularly located at both sites of linkage within polymer **1A** are ascribed to local changes of the tunnelling barrier via surface dipoles generated by the lone pair of $-\text{CN}$ groups [*Small* 14, 1704321 (2018).] and [*Nat. Mater.* 9, 320–323 (2010)]. Related depressions have previously been reported for other functional groups as well, featuring lone pairs such as pyridine [*Nat. Mater.* 9, 320–323 (2010)], the carboxylate group [*Nat. Mater.* 9, 320–323 (2010)] and deprotonated alkynes [*J. Phys. Chem. C* 119, 9669-9679 (2015).].

We have added the relevant description:

Line 133 “The depressions that are regularly located at both sites of linkage within polymer **1A** (indicated by white arrows) are ascribed to local changes of the tunnelling barrier via surface dipoles generated by the lone pair of $-\text{CN}$ groups^{16,17}.”

Comment 13

Line 124: A more detailed explanation is needed for why the $-\text{CH}_2-$ groups exhibit local

protrusions in the NC-AFM image.

Author reply:

The carbon atoms in the $-\text{CH}_2-$ groups exhibit sp^3 hybridization. When adsorbed on a metal surface, one C–H bond points towards the substrate, and another points towards the vacuum, resulting in a non-planar structural characteristic. Consequently, this unique arrangement manifests as a distinct bright spot feature in nc-AFM images [*Nat. Nanotech.* 12, 308–311 (2017)]. We have also made corresponding modifications in the manuscript as follows:

Line 125 “*The $-\text{CH}_2-$ groups exhibit local protrusions in the nc-AFM image because of the enhanced Pauli repulsion between the tip and non-planar bonding geometry. Due to the sp^3 -hybridization of the carbon atom in the $-\text{CH}_2-$ group, two H atoms connected to $\text{C}(sp^3)$ are oriented towards the substrate and the vacuum respectively, exhibiting a non-planar bonding geometry. This results in a reduced distance between the tip and the $-\text{CH}_2-$ group, specifically manifesting as bright protrusions in nc-AFM images* (white dotted circle Fig. 2d)¹⁵.”

Comment 14

Lines 178-180: The authors stated that “STM (Fig. 3a-c) and BR-STM (Fig. 3d-f) characterizations are employed to monitor the transitioning process of carbon hybridization.” However, this is not accurate; they cannot monitor the transitioning process and only show the structures of the intermediates.

Author reply:

We appreciate this suggestion from the reviewer. We have revised the corresponding description as follows:

Line 204: “*STM (Fig. 3a-c), nc-AFM (Fig. 3aII-cII) and BR-STM (Fig. 3d-f aIII-cIII) characterizations are employed to monitor the transitioning process of carbon hybridization characterize the corresponding products.*”

Comment 15

Line 190: The authors stated that the BR-STM image in Figure 3e unambiguously reveals the absence of the right-side $-\text{CN}$ group at the linkage. How did the authors conclude that the contrast around the connection site corresponds to the $-\text{CN}$ group?

Author reply:

Thanks for the reviewer's comments. Indeed, it is difficult to accurately determine the

characteristic of the –CN group solely through BR-STM characterization.

Fig. R4 | The identification of –CN group by nc-AFM measurement. a, nc-AFM image and corresponding chemical structure of polymer **1B**. **b,** nc-AFM image and corresponding chemical structure of oligomer functionalized with a –CN group.

Consequently, we have conducted a comparative analysis on the differences of cyano group in the nc-AFM imaging between our current work and previous studies. We can clearly find that the cyano group will exhibit a bifurcation feature at the end in the nc-AFM image of polymer **1B** (Fig. R4a), which is consistent with a previous work (Fig. R4b) [*Nat. Commun.* 11, 1490 (2020).].

We have supplemented the corresponding description as follows:

Line 141 “*The –CN groups exhibit a characteristic of terminal bifurcation in nc-AFM imaging, consistent with the results reported in previous work¹⁵.*”

Response to reviewer #2

Wei Xiong et al. report in this work a strategy to fabricate polymers taking advantage of tailor-made precursors functionalized with methylcyano ($-\text{CH}_2-\text{CN}$) groups on surfaces. The characterization of the resulting polymers is done by scanning tunneling microscopy and non-contact atomic force microscopy, which helped them to probe the transformation process across three distinct carbon-carbon bonds types: $\text{C}(\text{sp}^3) - \text{C}(\text{sp}^3)$ to $\text{C}(\text{sp}^2)=\text{C}(\text{sp}^2)$ and finally to $\text{C}(\text{sp}) \equiv \text{C}(\text{sp})$. I find this work interesting for the general reader of Nature Communications as it illustrates a novel pathway toward the regular formation of carbon-carbon bonds using functional groups unexplored so far on surfaces. The work is technically excellent, however, I consider that there are minor details that should be clarified by the authors before publication.

Comment 1

1. My main concern is about the way the authors describe the results in the title and abstract. I do not think that it is appropriate or accurate to define it as "precise manipulation". This term is mostly used in experiments where one gets an exquisite control of the chemical changes to be made, i.e. by tip manipulation. In this work it looks like the authors do a standard annealing process with the $\text{C}(\text{sp}^3)-\text{C}(\text{sp}^3)$ and $\text{C}(\text{sp}^2)=\text{C}(\text{sp}^2)$ states as intermediate steps of the final reaction. Therefore I strongly suggest to modify this is the title and related parts of the manuscript. I would actually include the used functional group in the title as I was expecting something completely different after reading it.

Author reply:

We appreciate this suggestion from the reviewer. Reference to our responses to the Reviewer #1 Comment 1, we have thoroughly reviewed our results and agree with the reviewer's assessment that the claims of "precise manipulation" and "high controllability" exceed the scope of our findings. Accordingly, we have toned down the related claims, including rephrasing the title, introduction, and related discussions in the main text. The relevant revisions are as follows:

The title is revised to be "*Visualizing stepwise evolution of carbon hybridization from sp^3 to sp^2 and to sp* "

Modifications to the abstract:

~~"Precise manipulation of carbon's three~~ *Regulating three carbon hybridization states lies at the heart of engineering carbon materials with tailored properties^{1,2}.*"

“..., ~~meticulously~~, orchestrating the sequential transition across ...”

“Surface science provides an exceptional platform for ~~the precise manipulation and visualization of controlling and visualizing~~ carbon hybridization at the atomic level by leveraging scanning probe microscopy^{7,10,11}.”

“Here, we ~~report a strategy to precisely manipulating carbon hybridizations in a controlled, stepwise manner, sequentially transitioning~~ present the visualization of stepwise evolution in carbon hybridizations, from sp^3 to sp^2 and ultimately to sp states...”

“..., we unequivocally probe the ~~thermally induced~~ transformation process across three distinct carbon-carbon bond types:...”

“~~Thermal annealing enables precise control over the entire reaction, conducting high yields of each bond type at corresponding steps.~~”

“Our work pioneers a breakthrough ... and serves as a dramatic advance ..., ~~enabling unprecedented controllability and flexibility~~ by employing novel cyanomethyl-substituted organic compounds.”

Modifications to the main text:

In several instances, the wording “manipulating” and “manipulation” has been replaced with “regulating” and “regulation”.

Line 57 “**Fig. 1 | Illustration of sequentially ~~manipulating~~ regulating carbon bond hybridization states by employing of the (4-methylcyano)phenyl-substituted compounds on surface.”**

Line 67 “The first and pivotal step towards ~~manipulating~~ regulating...”

Line 90 “(i) The $-CH_2-CN$ groups within **M1**, **M2** and **M3** allow the formation and ~~manipulation~~ regulation of carbon-carbon bonds...”

Line 307 “Notably, polymers **2B** and **2C** embedded with acrylonitrile and ethynylene units can be ~~selectively~~ synthesized with a ~~relatively high yield~~ selectivity on Ag(110) surface (Extended Data Fig. 5g-h).”

Line 311 “Additionally, distinct from..., our work offers a dehydrogenation-initiated, cleaner synthetic route to synthesizing ethynylene-bridged polymers ~~with exceptional quality and length (submicron level)~~ on Ag(110) surface.”

Modifications to the Conclusion:

“To conclude, we successfully ~~manipulate the carbon hybridization across all three states at the linkages within polymers fabricated by precursors containing $\text{CH}_2\text{-CN}$ groups on surfaces.~~ By using STM, BR-STM, and nc-AFM imaging measurements, we realize in-situ visualizations of the atomic configurations and the transformation process of as-formed polymers with distinct carbon linkages in real space by using STM, BR-STM, and nc-AFM imaging measurements, wherein the carbon hybridization evolves across all three states.”

“~~The characterizations, complemented by DFT calculations,~~ provide insights into the mechanism underlying the reaction pathways.”

“The electron-withdrawing -CN group... contributing to the ~~manipulation~~ evolution of...”

“The realization of ~~precisely manipulating~~ regulating carbon hybridization...”

“The proposed strategy represents a pioneering approach to ~~regulating~~ ~~manipulating~~ carbon hybridization...”

Comment 2

My second concern is related to the DFT-calculated energy barriers across the different configurations manuscript. I am not an expert in theoretical calculations, though it was surprising to me that the final state (FS) of all the reactions the authors present are higher in energy than the initial step (IS), which makes no sense to me. Can the authors comment on that?

Author reply:

We thank the review for pointing this out. The reviewer is correct in noting that, under equilibrium conditions, the energy of the products is generally lower than that of the reactants, with this energy decrease driving the reaction direction. However, in the STM chamber, we are dealing with non-equilibrium conditions. Specifically, the reactants undergo a chemical reaction, releasing H atoms (or H_2 molecules) and HCN molecules, which are immediately desorbed into the vacuum. The mechanical pump maintaining the vacuum in the chamber promptly removes these molecules, preventing them from returning, and this affects the directionality of the chemical reactions. A previous experiment, where HCN molecules were produced similarly to our work, also detected the leakage of HCN molecules from the chamber [*J. Phys. Chem. C* 121, 27521-27527 (2017)]. Thus, it is common for the energy of the product to be higher than that of the reactant in on-surface chemical reactions, as demonstrated by the example of methylene dehydrogenation on Au(110) [*Science* 334, 213-216 (2011)].

Response to reviewer #3

Response to reviewer #4

Xiong et al. reports a study on the precise manipulation of carbon hybridizations sequentially transitioning from sp^3 to sp^2 and ultimately to sp states at the linkages within polymers synthesized by several precursors with methylcyano ($-\text{CH}_2-\text{CN}$) groups on surfaces. By SPM and DFT measurements, they found that the $-\text{CN}$ group, which exhibits electron-withdrawing effect and easy dissociation at higher temperature, is the key factor for triggering dehydrogenation reaction (step I) and elimination reaction (step II/III). They further demonstrated the universal applicability of this strategy on 1D molecular wires as well as 2D COF on Ag(111) and Ag(110) surfaces respectively. The strategy overcomes the constraint that carbon hybridization has previously been confined to transitioning solely between two hybridization types and proposes a new type of on-surface reaction. The manuscript is well written, and the data presented are of high quality. Thus, I recommend its publication in Nature Communications after addressing the following questions:

Comment 1

In the description of Figure 2h (line 130-131), the author states “The sp^3 -hybridized L(C–C) is evidenced by the brighter contrast in the nc-AFM image.” Does this contrast difference in nc-AFM imaging also change after the elimination reaction of the $-\text{CN}$ group (step II)? Please provide the corresponding characterization results of the intermediated polymers, if available.

Author reply:

The brighter contrast of connection sites within polymer **1A** revealed by the nc-AFM imaging emerges as an important criterion in confirming the presence of $\text{C}(sp^3)-\text{C}(sp^3)$ bonds. According to the suggestion from reviewer, we have conducted more nc-AFM characterizations on some polymer intermediates undergoing partial elimination reactions, as shown in Supplementary Fig. 12. From the results, it can be observed that the brightness contrast at the connection site after the cyano elimination reaction significantly diminishes, which further substantiate this conclusion.

Supplementary Fig. 12 | High-resolution characterization of the intermediated polymer with linkages $L(C-C)$ and $L(C=C)$. a-d, High-resolution STM (200 mV, 100 pA) (a), BR-STM (2 mV, $\Delta z = 170$ pm) (b), nc-AFM (2 mV, $\Delta z = 200$ pm) (c) and Laplace-filtered images (d) of the polymer with both linkage $L(C-C)$ and $L(C=C)$.

In addition, we have also supplemented the manuscript with relevant content, as detailed below:
 Line 229 “Moreover, we have discerned significant alterations in the bright contrast of linkages emanating from the partially eliminated intermediate polymer. This observation reinforces the notion that structural planarization is indeed facilitated by the presence of sp^2 hybridized carbon atoms (Supplementary Fig. 12).”

Comment 2

As observed in Figure 2e and the supporting information provided by the authors, the M1 precursors undergo not only dehydrogenative polymerization but also crosslinking reactions upon annealing at 520 K on Au(111) surface. Can the authors explain the reasons behind this crosslinking reaction? And are there any other strategies that can be employed to reduce the yield of such crosslinking structures?

Author reply:

We appreciate this comment from the reviewer. Firstly, due to the chiral connectivity of M1 molecules within polymer 1A, the polymer tends to bend during its growth process. This bending influences its growth direction, making it more likely to encounter other chain structures. Notably, under high-temperature annealing conditions, when the cyano group comes

into close proximity with pyrene skeleton, a cyclization reaction between them is triggered, as shown in Fig. R5, thereby restricting the further growth of polymer **1A** [*Chem. Commun.* 55, 11611-11614 (2019).] [*Nat. Commun.*, 11, 1490 (2020)].

Fig. R5 | Cross-linked structure obtained via annealing M1 to 520 K on Au(111) surface. **a-b**, High-resolution STM (200 mV, 100 pA) images showing “T” and “Y” cross-linked by-products arising from cyclization reaction between cyano groups and pyrene skeletons. **a-b**, High-resolution STM images overlaid by corresponding chemical structures.

Secondly, the Au(111) surface exhibits a threefold symmetric anisotropy, which allows precursors to polymerize along multiple directions during the formation of 1D polymer chains. This, in turn, affects the final length of polymer **1A**, a phenomenon that has been commonly observed in previous study on the construction of polyarylene on Au(111) surfaces [*Nat. Synth.*, 1, 289–296 (2022)].

To reduce the cross-linked products in the reaction, we employ **M2** precursors to undergo the dehydrogenative homocoupling. Fig. R6 shows the highly ordered raft-like 1D polymers obtained via annealing **M2** molecules to 520 K for 20 mins on Au(111) surface. It is evident that there is a significant reduction in the by-products generated from cross-linked reactions.

Fig. R6 | Dehydrogenative polymerization of M2 on Au(111) surface. a-b, Overview STM images (200 mV, 100 pA) showing the self-assembled **M2** (a) and polymer **2A** (b) on Au(111) surface. c-d, Magnified STM images (200 mV, 100 pA) corresponding to (a-b). e, BR-STM (2 mV, $\Delta z = 170$ pm) images of the polymer **2A** with the linkages L(C-C). f, Yield statistics for the products resulting from the dehydrogenation reaction of **M2** molecules after annealing to 500 K and 520 K.

Comment 3

The elimination reaction induced by the dissociation of $-\text{CN}$ group is crucial for the transformation of connection sites within polymers. The authors need to clarify the type of elimination reaction in this work based on previous reports about on-surface elimination reactions [J. Am. Chem. Soc. 2023, 145, 11, 6203–6209].

Author reply:

We appreciate this valuable comment from the reviewer. After carefully reviewing Prof. Xu's work [J. Am. Chem. Soc. 145, 6203-6209 (2023).], we consider that the elimination reaction involving the dissociation of HCN group in our work should be assigned to the β -elimination reaction. Specifically, this involves the dissociation of the $-\text{CN}$ group on one carbon atom and the H atom on another carbon atom within the linkage site.

We have supplemented the description of the elimination reaction types in the manuscript as follows, and have revised Extended Data Fig. 3.

Line 221 “It is important to emphasize that the elimination reaction type in this work is assigned to β -elimination, which involves the dissociation of the $-\text{CN}$ group on one carbon atom and the H atom on another carbon atom within the linkage site²⁰. Two potential structures derived from α -elimination are discounted due to the absence fingerprints for radical electrons or methylene group in the experimental nc-AFM characterization result (Extended Data Fig.

3). *Two other possible products of step (II), which contain L(C-C) with radical electrons or saturated CH₂ groups at linkages, are excluded for nc-AFM images of polymer 1B, as no local protrusions or non-planar characteristics are observed.*”

Extended Data Fig. 3 | Identification of the chemical structures of the polymers 1B with the linkage L(C=C) elimination reaction types. *a-b*, STM (100 mV, 80 pA) nc-AFM images (2 mV, $\Delta z = 220$ pm) showing the homochiral and heterochiral polymers 1B with the linkage L(C=C). *c*, BR STM images (2 mV, $\Delta z = 190$ pm) corresponding to (*a-b*) respectively. *d-e*, nc-AFM image (2 mV, $\Delta z = 220$ pm) corresponding to (*a-b*) respectively. *f-h*, Laplace-filtered images of (*e-f*). *i*, The most probable chemical structures corresponding to (*a-b*) respectively. *j*, Two other possible chemical structures that are excluded. The upper and lower panels present different isomers of polymer 1B. Scale bars, 0.6 nm (*a-h*). *c*, Chemical structures of polymer 1B induced by β -elimination reaction. *d-e*, Two potential chemical structures of polymer 1B induced by α -elimination reaction.

Comment 4

The authors stated that 1D and 2D polymers containing L(C-C) linkages cannot be observed on the Ag(110) and Ag(111) surfaces, but the currently presented experimental results were not convincing enough to support this conclusion. Can the authors provide more detailed experimental results at lower temperatures that initially triggered the reaction on Ag(111) or Ag(110) surfaces?

Author reply:

We have repeated the experiment of annealing **M1** molecules to 420 K on Ag(111) surface, and the results are included as Supplementary Fig. 16. At this annealing temperature, in addition to some self-assembled structures, partial molecules undergo polymerization to form oligomers, confirming that this temperature is the initial polymerization temperature of **M1** molecules on Ag(111). The enlarged STM images show that the morphology of bonding site belongs to $L(C=C)$ (marked by red arrow), with no presence of $L(C-C)$. This observation supports the conclusion in the manuscript that “polymers containing $L(C-C)$ linkages cannot be observed on the Ag(110) and Ag(111) surfaces.”

Supplementary Fig. 16 | The formation of oligomers with the linkage $L(C=C)$ on Ag(111) surface via annealing **M1 to 420 K for 20 mins. a-b, Overview and magnified STM images (200 mV, 100 pA) showing a coexistence of self-assembled **M1** molecules and initial polymeric oligomers with the linkage $L(C=C)$ on Ag(111) surface.**

We have revised the relevant descriptions in the manuscript as follows:

Line 297 “The polymer **2A**, being supposed to contain $L(C-C)$, is not observed (Supplementary Fig. 16), which may be attributed to the high catalytic activity of Ag(110) and the lower energy barrier for the transformation of linkage from $L(C-C)$ to $L(C=C)$.”

Comment 5

Can the sequential transition of carbon hybridization also be achieved within COF structures on Au(111) surface?

Author reply:

We appreciate this comment from the reviewer. Reference to our responses to the Reviewer #1 Comment 7, we annealed **M3** to elevated temperatures on Au(111) surface, as shown in Supplementary Fig. 15. The experimental results confirm that the sequential transition of carbon hybridization can also be achieved with the utilization of **M3**.

Supplementary Fig. 15 | The transition of carbon hybridization within 2D COFs fabricated via annealing **M3 to different temperatures on Au(111). a-c, Large-scale STM images (200 mV, 150 pA) showing the COF structures fabricated by annealing **M3** to 520 K (a), 550 K (b), and 620 K (c) on Au(111), respectively. d-f, Magnified STM image (400 mV, 150 pA) derived from (a-c).**

Comment 6

Based on previous reports [ACS Nano 2017, 11, 7355–7361], –CN groups tend to undergo cyclization reactions with aromatic structures during the annealing process. Why does the cyano group exhibit a preference for dissociating from the linkage site and undergoing an elimination reaction, rather than engaging in a cyclization reaction with the adjacent phenyl group?

Author reply:

Upon reassessing the characterization data of the polymers obtained via dehydrogenation polymerization of **M1** molecules, we discovered no indications of cyclization reactions involving the cyano groups and their adjacent benzene rings. Given that only a limited number of cyano groups attached to the edges of graphene nanoribbons undergo ring-closure reactions at 620 K [*ACS Nano*, 11, 7355–7361 (2017)], it becomes evident that the 520 K temperature, which initiates the dehydrogenation polymerization, falls short of the threshold required for the cyano groups to engage in cyclization reaction.

Comment 7

Regarding reaction barriers, typically the product shall have less free energy. However, in the calculated energy path the final product has a relatively higher energy than the others in the path. Please give some comments.

Author reply:

The reviewer is correct in noting that, under equilibrium conditions, the energy of the products is generally lower than that of the reactants, with this energy decrease driving the reaction direction. However, in the STM chamber, we are dealing with non-equilibrium conditions. Specifically, the reactants undergo a chemical reaction, releasing H atoms (or H₂ molecules) and HCN molecules, which are immediately desorbed into the vacuum. The mechanical pump maintaining the vacuum in the chamber promptly removes these molecules, preventing them from returning, and this affects the directionality of the chemical reactions. A previous experiment, where HCN molecules were produced similarly to our work, also detected the leakage of HCN molecules from the chamber [*J. Phys. Chem. C* 121, 27521-27527 (2017)]. Thus, it is common for the energy of the product to be higher than that of the reactant in on-surface chemical reactions, as demonstrated by the example of methylene dehydrogenation on Au(110) [*Science* 334, 213-216 (2011)].

Comment 8

In calculations, are those small prototype molecules able to reasonably describe the energy barriers for the realistic big molecules?

Author reply:

Thank the reviewer for raising this concern. We have considered this issue when choosing the prototype molecules to calculate the dehydrogenation barriers. The critical point in our

considered dehydrogenation is the electron-withdraw nature of $-CN$ group. Thus our concern was whether or not in small molecules the electron-withdraw nature of $-CN$ group persists. We performed careful calculations on the projected density of states (PDOS) on both bigger molecule and smaller molecules as shown in Supplementary Fig. 6. The PDOS on each functional group, respectively, show identical features between two molecules, expect for a constant shift of the Fermi level, which is understandable for in bigger molecule the involved pyrene functional group will modify the Fermi level of the molecule. The influence of the electron-withdraw group $-CN$ on dehydrogenation process has been effectively simulated.

Supplementary Fig. 6 | The projected density of states (PDOS) on both MI molecule and phenylacetonitrile molecules. The PDOS on each functional group, respectively, show identical features between two molecules, expect for a constant shift of the fermi level, which is understandable for in bigger molecule the involved pyrene functional group will modify the fermi level of the molecule. The influence of the electron-withdraw group $-CN$ on dehydrogenation process has been effectively simulated.

The relevant descriptions have been added into the manuscript as follows:

Line 157 “The simplified model of phenylacetonitrile is believed to reasonably describe the energy barriers (Supplementary Fig. 6).”

Comment 9

Some minors: (1) The full name of BR-STM is needed; (2) Two Supplementary Figs. 13.

Author reply:

We appreciate these comments from the reviewer. We have revised the above issues in the manuscript.

Response to reviewer #5

A point-by-point response to the reviewers' comments

We are deeply grateful to all the reviewers for their acknowledgment of our revised manuscript. We have addressed the remaining comments from Reviewer 1# and revised the manuscript accordingly. In this response letter, comments from the reviewers are summarized in blue typeface. Our responses are in normal black font. The text in the manuscript and supporting information are presented in *italics* and the revised parts are highlighted with yellow shadow.

Response to reviewer #1

We (co-reviewers) appreciate the efforts that the authors addressed to our concerns in the initial manuscript. It is obvious that simulated dI/dV mappings based on the DFT calculations and the precise experiment on the DBP molecule and new precursor M4 improved the manuscript significantly. The authors also conducted a non-contact atomic force microscopy (nc-AFM) experiment on polymer 1C, providing compelling evidence of the bond transformation. However, they mention that achieving an orthohexagonal COF 3C structure is challenging. If this is the case, the schematic illustration of COF 3C in Figure 4g and the corresponding description may no longer be appropriate. We think that this point should be clarified in the manuscript before accepting this contribution for the publication.

Author reply:

We appreciate the reviewers' commendation of our revision efforts and agree that some more detailed description on COF 3C's morphology is helpful to avoid possible misleading. We added the following sentences to the main text.

Line 330 "*The as-formed COF 3C exhibits hexagonal units with linear edges of uniform length in its morphology (Figs. 4j and 4k). However, slight distortion is observed compared to the illustration in Fig. 4g, likely due to the strain retained by the flexible COF, which inhibits its complete relaxation.*"